# Enabling Realtime Reinforcement Learning at Scale with Staggered Asynchronous Inference

**Matthew Riemer**[*1,2,3]**, Gopeshh Subbaraj**[*1,2]**, Glen Berseth**[1,2]**, Irina Rish**[1,2]

[1]Mila, [2]Université de Montréal, and [3]IBM Research

## Abstract

Realtime environments change even as agents perform action inference and learning, thus requiring high interaction frequencies to effectively minimize regret. However, recent advances in machine learning involve larger neural networks with longer inference times, raising questions about their applicability in realtime systems where reaction time is crucial. We present an analysis of lower bounds on regret in realtime reinforcement learning (RL) environments to show that minimizing long-term regret is generally impossible within the typical sequential interaction and learning paradigm, but often becomes possible when sufficient asynchronous compute is available. We propose novel algorithms for staggering asynchronous inference processes to ensure that actions are taken at consistent time intervals, and demonstrate that use of models with high action inference times is only constrained by the environment's effective stochasticity over the inference horizon, and not by action frequency. Our analysis shows that the number of inference processes needed scales linearly with increasing inference times while enabling use of models that are multiple orders of magnitude larger than existing approaches when learning from a realtime simulation of Game Boy games such as Pokémon and Tetris.

## 1 Introduction

An often ignored discrepancy between the discrete-time RL framework and the real-world is the fact that the world continues to evolve even while agents are computing their actions. As a result, agents are limited in the types of problems that they can solve because the speed at which they can compute actions dictates a particular stochastic or deterministic time discretization rate. Agents that take infrequent actions require some lower-level program to manage behavior between actions, often through simple policies like remaining still or repeating the last action. Ideally, intelligent agents would exert more control over their environment, but this conflicts with the trend of using larger models, which have high action inference and learning times. Consequently, as typically deployed with sequential interaction, large models, which are often found to be essential for complex tasks, increasingly rely on low-level automation, reducing their control over realtime environments. This paper examines this discrepancy and explores alternative asynchronous interaction paradigms, enabling large models to act quickly and maintain greater control in high-frequency environments.

Figure 1a shows the standard sequential interaction paradigm of RL. In this setup, the agent receives a state from the environment, learns from the state transition, and then infers an action. Each process must be completed before the agent can process a new state, limiting the action frequency and increasing reliance on low-level automation as the model size grows. In contrast, Figure 1b illustrates the asynchronous multi-process interaction paradigm we propose. Our key insight is that even models with high inference times can act at every step using sufficiently many staggered inference processes. Similarly, sufficiently many asynchronous learning processes can maintain rapid updates without blocking progress, despite high learning times. This work formalizes and empirically tests the benefits and limitations of this approach, making the following contributions:

1. We formalize how the choice of a particular time discretization induces a new learning problem and how that problem relates to the original problem in Definition 1.

---

[*]Equal contribution. Direct correspondences to {matthew.riemer,gopeshh.subbaraj}@mila.quebec.

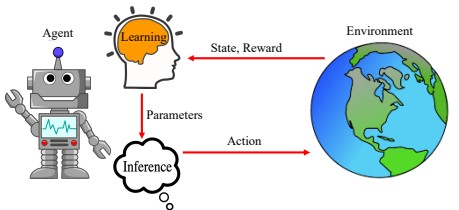
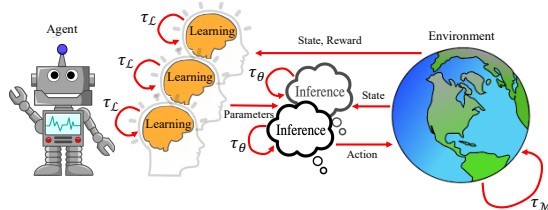

(a) Sequential Interaction and Learning      (b) Asynchronous Multi-process Interaction and Learning

Figure 1: **Frameworks for Environment Interaction in RL**. a) The typical sequential interaction paradigm where both learning and action inference block the environment from moving forward. b) The more realistic setting considered in this work where the environment, the agent's inference process, and agent's learning process all proceed at their own rate and interact asynchronously. Multiple self-loops are depicted to denote multiple asynchronous processes. $\tau_{\mathcal{M}}$ denotes the frequency of the environment, $\tau_{\theta}$ denotes the frequency of each inference process, and $\tau_{\mathcal{L}}$ denotes the frequency of each learning process. Sequential interaction and learning has a frequency of $\tau_{\mathcal{M}} + \tau_{\theta} + \tau_{\mathcal{L}}$.

2. We derive worst-case lower bounds on regret for solving the new problem rather than the original in Theorem 1, leading us to conclude in Remark 1 that typical sequential interaction (Figure 1a) scales poorly with model size.

3. We propose novel methods for staggering asynchronous inference in Algorithms 1 and 2, addressing the poor scaling properties of sequential interaction (Remark 2).

4. We conduct comprehensive experiments to verify our theory, demonstrating the use of models that are orders of magnitude larger for realtime games like Pokémon and Tetris.

## 2   REGRET DECOMPOSITION IN REALTIME REINFORCEMENT LEARNING

**Background - Sequential Interaction:** Most RL research focuses on agents interacting sequentially with a Markov Decision Process (MDP) [71; 94] $\mathcal{M}_{\text{seq}} = \langle \mathcal{S}, \mathcal{A}, p, r \rangle$, where $\mathcal{S}$ is a set of states, $\mathcal{A}$ is a set of actions, $r(s, a)$ is a reward function with outputs bounded by $r_{\max}$, and $p(s'|s, a)$ is a state transition probability function. Agents take actions based on a policy $\pi_{\theta}(a|s)$ that maps states to action probabilities parameterized by $\theta$. An unrealistic implicit assumption of this setting is that the time between decisions is fixed and only depends on the MDP. It is also unrealistically assumed that the environment can be paused while the policy generates an action $a$ from state $s$.

**Asynchronous Interaction Environments:** The standard MDP formalism lacks a crucial element for realtime settings where the environment cannot be "paused," and the agent interacts with it asynchronously, as described by Travnik et al. [102]. In this case, it is necessary to define the environment's behavior when the agent has not selected an action. We believe the most general solution is to use a preset default behavior if there is no available action $a_t$ by the agent $\pi$ at time-step $t$. This behavior follows $a \sim \beta(s)$, where $a \in \mathcal{A}_{\beta}$ is possibly from a different action space than $\mathcal{A}$, requiring $p$ and $r$ to be defined over $\mathcal{A} \cup \mathcal{A}_{\beta}$. Now we can define an asynchronous MDP $\mathcal{M}_{\text{async}} = \langle \mathcal{S}, \mathcal{A}, p, r, \beta \rangle$ as an extension of a sequential MDP $\mathcal{M}_{\text{seq}}$ with the addition of the default behavior policy $\beta$. Note that $\beta$ does not need to be non-Markovian, because the state space should be defined to include any intermediate computations needed to generate the actions of $\beta$. Defining the default behavior as a policy is no more than a useful interpretation of what happens and is equivalent to saying the environment follows a Markov chain $p^{\beta}(s'|s)$ when no action is available where $p^{\beta}(s'|s) := \sum_{a \in \mathcal{A}_{\beta}} p(s'|s, a)\beta(a|s)$ with expected reward $r^{\beta}(s) = \sum_{a \in \mathcal{A}_{\beta}} \beta(a|s)r(s, a)$.

**Time Discretization Rates:** The real environment evolves in continuous time, so we must define time discretization rates to describe each component of the agent-environment interface in discrete steps. We treat the environment step time as a random variable $\text{T}_{\mathcal{M}}$ with sampled values $\tau_{\mathcal{M}} \sim \text{T}_{\mathcal{M}}$ and expected value $\bar{\tau}_{\mathcal{M}} := \mathbb{E}[\text{T}_{\mathcal{M}}]$. Similarly, the inference time of the policy for a single action[1] is another random variable $\text{T}_{\theta}$ with sampled values $\tau_{\theta} \sim \text{T}_{\theta}$ and expected value $\bar{\tau}_{\theta} := \mathbb{E}[\text{T}_{\theta}]$.[2] We can

---

[1]See Appendix B.5 for a comparison with *action chunking* methods that produce multiple actions at a time.
[2]While policies in general could have adaptive computation times based on the state, this is relatively uncommon in the literature and will be left to future work for simplicity of the discourse.

now introduce yet another random variable $T_{\mathcal{I}}$ with sampled values $\tau_{\mathcal{I}} \sim T_{\mathcal{I}}$ and expected value $\bar{\tau}_{\mathcal{I}} := \mathbb{E}[T_{\mathcal{I}}]$ that is of particular importance to our work, representing the time elapsed between actions taken by the agent $\pi_\theta$ (rather than $\beta$) in $\mathcal{M}_{\text{async}}$. This has been called a variety of names in the literature including the interaction time, the action cycle time, and the inverse of the interaction frequency. What is very important to note for our purposes is that $T_{\mathcal{I}}$ need not be equal to $T_{\mathcal{M}}$ nor $T_\theta$, with the precise relation between these variables depending on the particular method of agent deployment. Establishing these three random variables now allows us to define the decision problem induced by these choices related to the agent-environment boundary (see Figure 2 for an illustration).

> **Definition 1 (Induced Delayed Semi-MDP)** *Any choice of random variables $T_{\mathcal{M}}$, $T_{\mathcal{I}}$, and $T_\theta$ applied to an asynchronous MDP $\mathcal{M}_{async}$ induces a delayed semi-MDP $\tilde{\mathcal{M}}_{delay} := \langle \mathcal{S}, \mathcal{A}, p, r, \beta, T_{\mathcal{M}}, T_{\mathcal{I}}, T_\theta \rangle$ where the semi-MDP decision making steps $\tilde{t}$ associated with the actual decisions of the agent $\pi$ happen after $\lceil \tau_{\mathcal{I}}/\tau_{\mathcal{M}} \rceil$ steps $t$ in the ground asynchronous MDP $\mathcal{M}_{async}$. The semi-MDP is delayed with respect to $\mathcal{M}_{async}$ because semi-MDP actions $\tilde{a}_{\tilde{t}} \in \mathcal{A}$ generated by $\pi$ are equivalent to actions that are delayed by $\lceil \tau_\theta/\tau_{\mathcal{M}} \rceil$ in $\mathcal{M}_{async}$ such that $\pi_\theta(\tilde{a}_{\tilde{t}}|s_{\tilde{t}}) = \pi_\theta(a_{t+\lceil \tau_\theta/\tau_{\mathcal{M}} \rceil}|s_t)$ where $s_{\tilde{t}} = s_t$. If $\lceil \tau_\theta/\tau_{\mathcal{M}} \rceil > 1$ the transition dynamics are $p^\beta$ and reward dynamics are $r^\beta$ for $\lceil \tau_\theta/\tau_{\mathcal{M}} \rceil - 1$ steps in $\mathcal{M}_{async}$ until $a_{t+\lceil \tau_\theta/\tau_{\mathcal{M}} \rceil}$ is applied.*

In general, the optimal policy and optimal reward rate will not be the same for $\mathcal{M}_{\text{async}}$ and $\tilde{\mathcal{M}}_{\text{delay}}$, with $\tilde{\mathcal{M}}_{\text{delay}}$ incurring additional sub-optimality because of the coarse nature of the decision problem. That said, we have direct control over $T_{\mathcal{I}}$ and $T_\theta$, so it is of interest to understand how our design decisions relate to the sub-optimality experienced. Chiefly, we are interested in understanding under what scenarios the optimal reward rate of $\mathcal{M}_{\text{async}}$ can still be achieved even when $\bar{\tau}_\theta >> \bar{\tau}_{\mathcal{M}}$. To do this, we focus on worst case lower bounds on regret i.e. the unavoidable regret incurred because of the interaction defined by $\tilde{\mathcal{M}}_{\text{delay}}$ in the worst case scenario where $\beta$ is always a suboptimal choice. The realtime regret $\Delta_{\text{realtime}}(\tau)$ is the accumulated suboptimality in $\tau$ seconds relative to following the optimal policy at every discrete steps $t$ occurring after $\tau_{\mathcal{M}}$ seconds in $\mathcal{M}_{\text{async}}$.

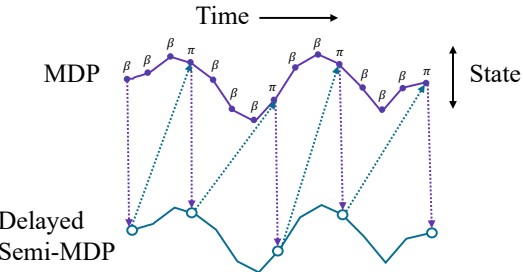

Figure 2: **Induced Delayed Semi-MDP.** We illustrate the semi-MDP described in Definition 1 following the style of Figure 1 from Sutton et al. [95]. $\mathcal{M}_{\text{async}}$ is depicted in purple and $\tilde{\mathcal{M}}_{\text{delay}}$ is depicted in blue. Actions are delayed by the inference time of the policy $\pi$ and the default policy $\beta$ is followed between selections.

> **Theorem 1 (Realtime Regret Decomposition)** *The accumulated realtime regret $\Delta_{realtime}(\tau)$ over time $\tau$ of a delayed semi-MDP $\tilde{\mathcal{M}}_{delay}$ relative to the oracle policy in the underlying asynchronous MDP $\mathcal{M}_{async}$ can be decomposed into three independent terms.*
>
> $$\Delta_{realtime}(\tau) = \Delta_{learn}(\tau) + \Delta_{inaction}(\tau) + \Delta_{delay}(\tau) \tag{1}$$
>
> $\Delta_{learn}(\tau)$ *is the regret experienced even in sequential environments as a result of learning and exploration. The lower bound in the worst case is:*[3]
>
> $$\Delta_{learn}(\tau) \in \Omega(\sqrt{\tau/\bar{\tau}_{\mathcal{I}}}) \tag{2}$$
>
> $\Delta_{inaction}(\tau)$ *expresses the regret as a result of following $\beta$ rather than optimal actions in $\mathcal{M}_{async}$. The lower bound and upper bound in the worst case is:*
>
> $$\Delta_{inaction}(\tau) \in \Theta((\tau/\bar{\tau}_{\mathcal{I}}) \times (\bar{\tau}_{\mathcal{I}} - \bar{\tau}_{\mathcal{M}})/\bar{\tau}_{\mathcal{M}}) \tag{3}$$
>
> $\Delta_{delay}(\tau)$ *expresses the regret as a result of the delay of actions by $\pi$ in the underlying asynchronous $\mathcal{M}_{async}$. The lower bound in the worst case is:*
>
> $$\Delta_{delay}(\tau) \in \Omega((\tau/\bar{\tau}_{\mathcal{I}}) \times \mathbb{E}[1 - (p_{minimax})^{\lceil \tau_\theta/\tau_{\mathcal{M}} \rceil}]) \tag{4}$$

> *where $p_{minimax} := \min_{s \in \mathcal{S}, a \in \mathcal{A}} \max_{s' \in \mathcal{S}} p(s'|s,a)$ is a measure of environment stochasticity[4] and $\lceil \tau_\theta / \tau_\mathcal{M} \rceil$ is the number of environment steps elapsed during action inference.*

See Appendix B for a formal proof of Theorem 1 and our other findings. We believe this work is the first to formally state the regret decomposition in Equation 1. Note though that previous studies on real-world RL have highlighted the challenges of learning from limited samples, realtime inference, and managing system delays in scaling methods to realtime settings [19]. Equation 2 extends known lower bounds on learning time [31], using the notation from Definition 1 to explicitly connect with continuous time. Notably, this bound depends on $\bar{\tau}_\mathcal{I}$ (not $\bar{\tau}_\theta$) and assumes learning can keep pace with the environment to learn from every interaction. Equation 3 provides a novel regret bound, formalizing the known suboptimality of interacting with realtime environments at a slower pace [102; 28; 73; 107; 106; 21]. This result highlights limitations of the sequential interaction paradigm.

> **Remark 1 (Inaction of Sequential Interaction)** *When $\pi$ and $\mathcal{M}_{async}$ interact sequentially, $\tau_\mathcal{I} = \tau_\theta$ such that in the worst case $\Delta_{inaction}(\tau) \in \Omega(\tau/\bar{\tau}_\theta \times (\bar{\tau}_\theta - \bar{\tau}_\mathcal{M})/\bar{\tau}_\mathcal{M})$. This implies that even as $\tau \to \infty$, in the worst case $\Delta_{realtime}(\tau)/\tau \in \Omega(\Delta_{inaction}(\tau)/\tau) \in \Omega((\bar{\tau}_\theta - \bar{\tau}_\mathcal{M})/\bar{\tau}_\mathcal{M}\bar{\tau}_\theta)$.*

This means a realtime framework with sequential interaction cannot ensure that regret will eventually dissipate. Thus, we explore asynchronous alternatives in the next section. Finally, Equation 4 highlights the key limitation in minimizing regret using asynchronous compute. Previous work established that suboptimality from delay in MDPs relates to the stochasticity in the underlying undelayed MDP [17; 64], focusing on communication delays inherent to the environment. Our focus, however, is on delays caused by the agent's computations, which we can control. Thus, the emphasis on regret associated with the decision that leads to a particular value of $\tau_\theta$ is novel. Since this term is the only part of regret that depends on $\tau_\theta$, it helps identify which environments are manageable when $\tau_\theta >> \tau_\mathcal{M}$. In deterministic environments, there is no regret due to $\tau_\theta$ as $p_{minimax} = 1$, but in stochastic environments, the degree and temporal horizon of stochasticity determine what values of $\tau_\theta$ are tolerable. For simplicity, we present a looser bound here; a tighter bound is available in Appendix B. In summary, stochasticity with respect to actual rewards is what really matters.

## 3 ASYNCHRONOUS INTERACTION & LEARNING METHODS

Figure 3 highlights key differences between the standard sequential RL framework and the asynchronous multi-process framework we propose. In the sequential framework, interaction and learning delay each other. In contrast, in the asynchronous framework that we propose, actions and learning can occur at every step with enough processes. However, actions are delayed and reflect past states, which may limit performance in some environments. Note that staggering processes to maintain regular intervals is essential. For example, if all inference processes took a deterministic amount of time with no offset between them, all additional actions in the environment would be overwritten with no benefit from increasing compute. Meanwhile, with staggering we can experience linear speedups.

### 3.1 BACKGROUND: STAGGERED ASYNCHRONOUS LEARNING

**Parallel vs. Asynchronous Updates:** Learning from a transition, i.e., computing gradients, usually takes longer than inference. Thus, performing learning in separate processes is crucial to avoid blocking inference [106], especially for models with a large number of parameters. For this use case, one might be tempted to consider parallel learning processes to increase the effective batch size without increasing wall-clock time per batch as this avoids wasted computation. Indeed, parallel updates are better for training large language models when final performance and compute efficiency are most important. In contrast, asynchronous learning can produce updates even faster than learning from a single transition, making the model more responsive to exploration. However, lock-free asynchronous approaches risk overwriting updates, potentially wasting computation that does not contribute to final performance. That said, our focus is on maximizing responsiveness in large models, not necessarily compute efficiency, and overwritten updates are not wasted with respect to regret.

---

[3] Known algorithms achieve regret upper bounds within a logarithmic factor of this lower bound [68].

[4] When the environment is deterministic, $p_{minimax} = 1$ and $\Delta_{delay}(\tau) = 0$. When the environment is uniformly random, $p_{minimax} = 1/|\mathcal{S}|$ and as $|\mathcal{S}| \to \infty$, $\Delta_{delay}(\tau) \in \Omega(\tau)$.

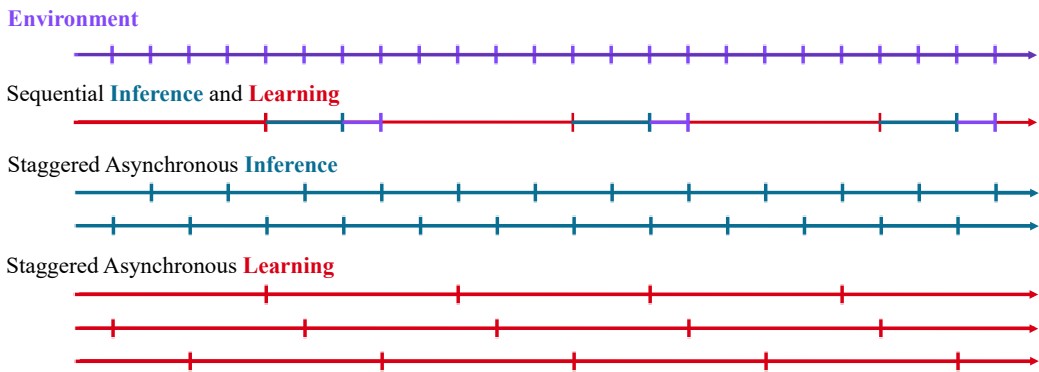

Figure 3: **Realtime Interaction Frequency.** We illustrate the comparative interaction frequency of methods that sequence learning and inference and those that maintain multiple staggered asynchronous processes. Even when inference times are greater than the environment step time, it is possible to use asynchronous compute to eliminate inaction and learn from every step.

**Round-Robin Asynchronous Learning:** Langford et al. [50] laid the foundation for addressing asynchronous update staggering for large neural network models using variants of stochastic gradient descent (SGD). They showed that applying updates in a delayed, orderly fashion avoids wasted compute on overwritten gradients. Their approach demonstrated convergence for delayed SGD, with linear scaling limited only by the time taken to update parameters relative to computing gradients. This method allows significant linear scaling with minimal compute waste for large models and the delay in the updates will not be a persistent source of regret in Theorem 1. While not our novel contribution, this strategy is underexplored. We investigate its scaling properties in our experiments.

### 3.2 OUR NOVELTY: STAGGERED ASYNCHRONOUS INFERENCE

In Remark 1 we highlighted that $\bar{\tau}_{\mathcal{I}}$ is fundamentally limited by $\bar{\tau}_\theta$ for sequential interaction, which results in persistent regret even as time goes on when $\bar{\tau}_\theta > \bar{\tau}_{\mathcal{M}}$. We will now highlight two novel algorithms for staggering inference processes that can lead to a reduction in $\bar{\tau}_{\mathcal{I}}$ when the number of inference processes $N_{\mathcal{I}}$ are increased. Algorithm 1 is capable of scaling the expected interaction time with the number of processes by $\bar{\tau}_{\mathcal{I}} \leq \min(\tau_\theta^{\max}/N_{\mathcal{I}}, \bar{\tau}_{\mathcal{M}})$ where $\tau_\theta^{\max}$ is the maximum encountered value of $\tau_\theta$. Meanwhile, Algorithm 2 is capable of scaling the expected interaction time with the number of processes by $\bar{\tau}_{\mathcal{I}} = \min(\bar{\tau}_\theta/N_{\mathcal{I}}, \bar{\tau}_{\mathcal{M}})$. Both algorithms can eliminate inaction.[5]

> **Remark 2 (Inaction of Asynchronous Interaction)** *For any $\bar{\tau}_\theta$ when $\pi$ and $\mathcal{M}_{async}$ interact asynchronously with staggering algorithms 1 or 2, there is a value of the number of inference processes $N_{\mathcal{I}}^*$ such that for all $N_{\mathcal{I}} \geq N_{\mathcal{I}}^*$, $\Delta_{inaction}(\tau)/\tau \to 0$ as time goes to $\tau \to \infty$.*

Algorithm 1 always ensures each processes waits for the current estimate of $\tau_\theta^{\max}$ amount of seconds before an action is taken by that process to preserve the spacing between actions. Adjustments are made to the waiting time in each process considering dist($x$,$y$), the distance process $x$ is behind process $y$ in the cycle of processes, until the estimate converges to the true $\tau_\theta^{\max}$ value. The benefit of this algorithm is that the spacing between actions stays very consistent with no variance once the maximum value estimate has stabilized. This makes $\tilde{\mathcal{M}}_{\text{delay}}$ easier to learn from. The downside is that the amount of necessary compute to eliminate inaction may be relatively high.

On the other hand, Algorithm 2 stops all waiting in all processes as time goes on, so that the expected interaction time of each process is $\bar{\tau}_\theta$. An estimate of $\bar{\tau}_\theta$ is maintained and when the estimate changes after an action is taken, processes wait for an amount of time designed to adjust the average spacing between processes to $\bar{\tau}_\theta/N_{\mathcal{I}}$. The law of large numbers ensures that the estimate converges to $\bar{\tau}_\theta$ in the limit as $\tau \to \infty$ and that the waiting time diminishes to zero. Algorithm 2 has a strictly smaller compute requirement than Algorithm 1, but experiences variance in $T_{\mathcal{I}}$ driven by the variance in $T_\theta$, which makes $\tilde{\mathcal{M}}_{\text{delay}}$ harder to learn from. The compute advantage becomes more significant for

---

[5]Appendix B.6 describes how this can lead to achieving sublinear regret in deterministic environments.

distributions that have variance in $T_\theta$ such that $\tau_\theta^{\max} - \bar{\tau}_\theta$ is large. In our experiments, we consider Algorithm 1 because we found the variance in $T_\theta$ is small for the models and hardware we consider.

---

**Algorithm 1** Maximum Time Inference Staggering

---

**Initialize:** $\hat{\tau}_\theta^{\max} = 0$ and delay[processnum] $= 0 \; \forall$ processnum $\in [1, ..., N_\mathcal{I}]$
**Run:** INFERENCE[processnum] $\forall$ processnum $\in [1, ..., N_\mathcal{I}]$
  **function** INFERENCE(processnum)
    **while** alive **do**
      sleep(delay[processnum])         $\triangleright$ Sleep for any delays accumulated by other processes
      delay[processnum] $\leftarrow 0$
      $a, \tau_\theta \sim \pi_\theta(s_t)$         $\triangleright$ Have the policy sample an action and inference time
      **if** $\tau_\theta \geq \hat{\tau}_\theta^{\max}$ **then**
        $\delta\tau \leftarrow \tau_\theta - \hat{\tau}_\theta^{\max}$     $\triangleright$ Other processes sleep for the difference with the maximum
        **for** num $\neq$ processnum $\in [1, ..., N_\mathcal{I}]$ **do**
          delay[num] $\leftarrow$ delay[num] + dist(num,processnum) $\times \delta\tau / N_\mathcal{I}$
        $\hat{\tau}_\theta^{\max} \leftarrow \tau_\theta$         $\triangleright$ Set new global maximum
      **else**
        sleep($\hat{\tau}_\theta^{\max} - \tau_\theta$)         $\triangleright$ Sleep for the remaining time
      $a_{t + \lceil \hat{\tau}_\theta^{\max} / \tau_\mathcal{M} \rceil} \leftarrow a$         $\triangleright$ Register action in environment

---

**Hardware Optimization:** In this paper, our focus is on the possibility of achieving speedups when adequate hardware is available to facilitate it. As such, we focus on what can be achieved with an ideal set of hardware rather than the most efficient way to utilize a given constrained set of hardware. In our experiments, we run each process on its own dedicated CPU such that resource constraints like memory capacity, and memory bandwidth do not present significant issues. However, if, for example, we aimed to implement multiple simultaneous processes on a single GPU, we would have to consider tradeoffs between memory capacity, memory bandwidth, and latency that will jointly serve to limit the possible speedups. We leave analysis of these practical tradeoffs to future work.

## 4 RELATED WORK

**Realtime interaction:** Previous work such as Travnik et al. [102] has considered the asynchronous nature of realtime environments. However, we are not aware of any prior paper that has formalized the connection between asynchronous and sequential versions of the same environment as we have. Travnik et al. [102] highlight the reaction time benefit of acting before you learn, and Ramstedt & Pal [73] highlight the reaction time benefit of interacting based on a one-step lag. Meanwhile, the interaction frequency of both of these approaches are limited by sequential interaction paradigm and thus the drawback highlighted in Remark 1 also applies to them.

**Designing the interaction rate:** Farrahi & Mahmood [21] examined how the choice of $\tau_\mathcal{I}$ affects the learning performance of deep RL algorithms in robotics. They found that low $\tau_\mathcal{I}$ complicates credit assignment, while high $\tau_\mathcal{I}$ complicates learning reactive policies. Karimi et al. [36] proposed a policy that executes multi-step actions with a learned $\tau_\mathcal{I}$ within the options framework, which may aid in slow problems where credit assignment is challenging. However, this approach does not address the action delay issue we focus on and may worsen it by committing to multiple actions based on a delayed state. Our policy, defined in the semi-MDP framework (Definition 1), relies on a low-level policy $\beta$, similar to the options framework [95]. The key difference is that $\beta$ cannot be modified, preventing intra-option learning and thus making it impossible to improve $\beta$ even when it is sub-optimal. Thus we would rather minimize the use of $\beta$ i.e. minimize inaction.

**Reinforcement learning with delays:** Reinforcement learning in environments with delayed states, observations, and actions is well-studied. Typically, delays are treated as communication delays inherent to the environment [103; 9]. In contrast, we focus on delays resulting from our computations, which are under our control and part of agent design. Our formulation of delay as part of regret is novel due to this unique focus. Common methods address delay by augmenting the state space with all actions taken since the delayed state or observation [8; 37; 63], but this is infeasible for us since these actions are not available when computation begins. Instead, our approach aligns more with

methods addressing delay without state augmentation [89; 9; 16; 2; 35]. However, these methods are limited by the environment's stochasticity [17; 64], as highlighted by Equation 4 of Theorem 1.

**Asynchronous learning:** Most work on asynchronous RL involves multiple environment simulators learned from asynchronously or in parallel [62; 20; 90]. We explore a more challenging real-world setup with a single environment, limiting exploration opportunities. Unlike typical asynchronous setups where each process interacts sequentially with the environment and then learn from that interaction [62], our setting benefits from making interaction and learning asynchronous (Remark 2). Indeed, our paper introduces the concept of staggering asynchronous interaction, which is an innovation with benefits unique to the non-pausing environment setting we explore. Similarly to ours, some prior work has considered asynchronous learning to avoid blocking inference [106], focusing on model-based learning [28; 29; 27; 107] and auxiliary value functions [96; 14]. The novelty of our approach is in its use of multiple asynchronous staggered inference processes instead of a single process, a critical contribution for deploying large models (see Remark 1 and Remark 2).

# 5 EMPIRICAL RESULTS

To show that our proposed method does indeed provide practical benefits for minimizing regret per second with large neural networks in realtime environments, we perform a suite of experiments to validate the theoretical claims made in the previous sections. Our experiments include:

- **Question 1:** an evaluation of the speed of progress through a realtime game strategy game where constant learning is necessary to move forward when action inference times are large.

- **Question 2:** an evaluation of episodic reward in games where reaction time must be fast to demonstrate that asynchronous interaction can maintain performance with models that are multiple orders of magnitude larger than those using sequential interaction.

- **Question 3:** an evaluation of the scaling properties of Algorithms 1 and 2 to demonstrate that the needed number of processes to eliminate inaction, $N_{\mathcal{I}}^*$, scales linearly with increasing inference times $\bar{\tau}_\theta$ and parameter counts $|\theta|$.

- **Question 4:** an evaluation of the scaling properties of round-robin asynchronous learning [50] to demonstrate that the number of processes needed to learn from every transition also scales linearly with increasing learning times and parameter counts $|\theta|$.

**Implementation Details:** For our experiments, we implemented the Deep Q-Network (DQN) learning algorithm [61] within our asynchronous multi-process framework, using a discount factor of $\gamma = 0.99$, a batch size of 16, and an Adam learning rate of 0.001 for the sparse reward Game Boy games (with 0.00001 used for the comparatively dense reward Atari games). A shared experience replay buffer stores the 1 million most recent environment transitions. For preprocessing, we down-sampled monochromatic Game Boy images to 84x84x1, similar to the preprocessing that is standard for Atari [61]. Following the scaling procedure previously established by [13] and [91], we used a 15-layer ResNet model [20] while scaling the number of filters by a factor $k$ to grow the network. The model sizes correspond to: $k = 1$ (1M parameters), $k = 7$ (10M), $k = 29$ (100M), and $k = 98$ (1B). Models were deployed on multi-process CPUs using Pytorch multiprocessing library on Intel Gold 6148 Skylake cores at 2.4GHz, with one core per process and multiple machines for models using $> 40$ processes. See Appendix A for further detail regarding our experimental setup and a discussion of limitations. As large models are known to be difficult to optimize in an online RL context, we also tried training common 1M models with an equivalent amount of added delay as the larger models to perform a sanity check. Our initial experiments indicated performance was similar.

**Realtime Environment Simulation:** To run a comprehensive set of scaling experiments that would not be feasible with real-world deployment, we need a simulation of a realistic realtime scenario. Towards this end, we considered two games from the Game Boy that are made available for simulation as RL environments through the Gymnasium Retro project [65]. We implemented a realtime version of the Game Boy where it is run at 59.7275 frames per second such that $\tau_{\mathcal{M}} = 1/59.7275$ and with "noop" actions executed as the default behavior $\beta$. This exactly mimics the way that humans would interact with the Game Boy as a handheld console [104] and matches the setting in which humans compete over speed runs for these games. We also leveraged three Atari environments [7] run at 60 frames per second with "noop" actions executed as $\beta$ to mimic the way that humans interact with the Atari console in the real-world. These are ideal settings for addressing our core empirical questions.

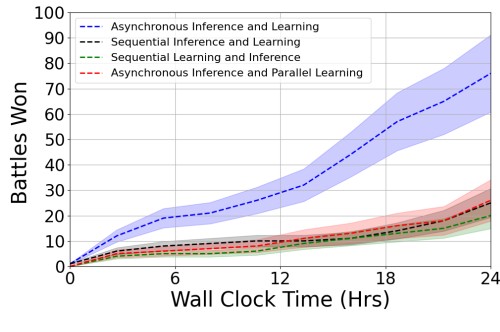
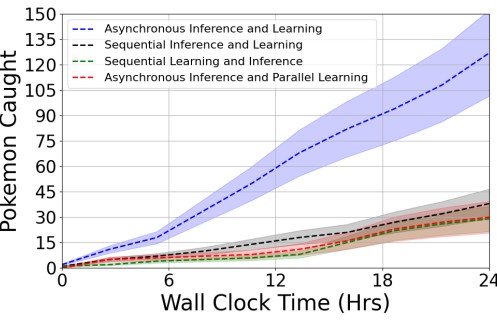

(a) 100M: Pokémon Battles Won vs. Time $\tau$       (b) 100M: Wild Pokémon Caught vs. Time $\tau$

Figure 4: **Realtime Pokémon Performance**. **a)** Battles won in Pokémon Blue over time for $|\theta| = 100M$. **b)** Wild Pokémon caught in Pokémon Blue over time for $|\theta| = 100M$. The parallel learning baseline considers an effective batch size that is $33\times$ larger, incurring $33\times$ fewer updates.

### 5.1 FASTER PROGRESS THROUGH A REALTIME GAME WITH CONSTANT NOVELTY

**Pokémon Blue:** Pokémon Blue is a valuable environment for our study due to its long play through time and constant novelty over many hours of play. Acting quickly is not a necessity to complete this game as it lets the agent dictate the pace of play, but better players are still differentiated based on their speed of completing the game. Indeed, the game has a large community of "speed runners" aiming to complete milestones in record times, with even the fastest milestones taking multiple hours [93]. It is an interesting domain for our study because acting quickly is only beneficial to the extent that the agent displays competent behavior, so action throughput alone will not lead to better results when the quality of play correspondingly suffers. Because Pokémon Blue is known as a challenging exploration problem that perhaps even exceeds the scope of previous deep RL achievements [30], we divided the game into two settings based on expert human play: 295 battle encounters (Figure 9a) and 93 catching encounters (Figure 9b). Agents are deployed in these settings and must complete each encounter (by winning a battle or catching a Pokémon) before progressing.

**Question 1:** *Can asynchronous approaches achieve faster progress in a realtime strategy game where constant learning is necessary to move forward even when action inference times are large?*

**Figure 4:** For Pokémon Blue we leverage $N_{\mathcal{I}} = N_{\mathcal{I}}^*$ and enough learning threads to learn every 5 environment steps. We did not find benefit from learning every step given that the underlying game is not responsive to every action taken at the frame level. For all models, the exploration rate is annealed from $1.0$ to $0.05$ over the course of the first 100,000 steps of learning. We compare to the standard RL interaction paradigm where inference and learning are performed sequentially [94] and when the order is flipped for realtime settings [102]. Our results in Figures 4a and 4b showcase that asynchronous inference and learning combine for superior scaling of realtime performance as models grow. See Appendix A for results for the 1M and 10M models. The improvement over sequential interaction corresponds with our expectations given Remarks 1 and 2.

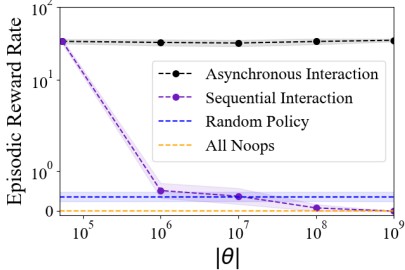

Figure 5: **Realtime Tetris Performance vs. $|\theta|$.** The average episodic return over 2,000 episodes of learning. We compare models with a single inference process to those that perform staggered asynchronous inference following Algorithm 1.

### 5.2 MAINTAINING PERFORMANCE IN GAMES THAT PRIORITIZE REACTION TIME

**Tetris:** We also explore the game Tetris (Figure 9c) that presents a different kind of challenge for our agents where even more of a premium is put on reaction time. In Tetris, the player will lose the game if they wait indefinitely and do not act in time. While a slow policy can eventually win in Pokémon, despite taking longer than necessary, a policy that does not act timely cannot progress in Tetris as new pieces must be moved correctly before they fall on existing pieces.

**Question 2:** *Can asynchronous interaction help for games that prioritize reaction time as $|\theta|$ grows?*

**Figure 5:** To aid with exploration and jump-start learning, a single episode of human play is provided to each agent to learn from. The agent continues to learn from a total of 2,000 episodes with an exploration rate of 0.05. We see that sequential interaction scales quite poorly for games that prioritize a high frequency of actions and cannot surpass random performance for $|\theta| \geq$ 1M as we would expect based on Remark 1. Meanwhile, staggered asynchronous inference following Algorithm 1 can achieve a much higher reward rate for $|\theta| >$ 1B as we would anticipate based on Remark 2.

**Figure 6:** We also provide supplemental results for three Atari games with 2,000 training episodes in the same setting (see Figure 10), demonstrating superior performance as $|\theta|$ scales up. Here we additionally compare to a deeper ResNet architecture with 30 convolutional layers rather than 15. The deeper architecture brings up inference times at the same number of total parameters and thus leads to an even bigger gap between staggered asynchronous and sequential approaches. We find that the ability for performance to be maintained with asynchronous methods as the model size grows depends on the game. For example, Boxing includes a very stochastic opponent policy, making it very hard to maintain human-level performance in the presence of delay, while the Krull environment is predictable enough for agents to surpass human performance even when delay is significant. In Figure 14 we see a similar overall trend while using Rainbow learning [26] rather than DQN learning.

## 5.3 COMPUTATIONAL SCALING OF ASYNCHRONOUS INTERACTION AND LEARNING

**Question 3:** *How does $N_{\mathcal{I}}^*$ from Remark 2 scale with $\bar{\tau}_\theta$ and the number of parameters $|\theta|$?*

**Figure 7:** We measure $N_{\mathcal{I}}^*$ for Algorithms 1 and 2 when the Game Boy is run at the standard frequency using an $\epsilon$-greedy DQN policy at $\epsilon = 0$ and $\epsilon = 0.5$. Figure 7a shows that $N_{\mathcal{I}}^*$ scales roughly linearly with $\bar{\tau}_\theta$ in all cases, as expected for effective staggering (Remark 2). Figure 7b also demonstrates that $N_{\mathcal{I}}^*$ scales roughly linearly with $|\theta|$. When $\epsilon = 0$, the variance in $\mathrm{T}_\theta$ is very small and Algorithms 1 and 2 thus perform the same. When $\epsilon = 0.5$, the variance in $\mathrm{T}_\theta$ is high because sampling random actions is very fast, showcasing predictably superior performance for Algorithm 2. The performance of Algorithm 1 is unaffected by stochasticity in $\mathrm{T}_\theta$, but the values of $\bar{\tau}_\theta$ are.

**Notation for Asynchronous Learning:** We also would like to consider the compute scaling properties of round-robin asynchronous learning [50]. We now assume that the time to learn from an environment transition can be treated as a random variable $\mathrm{T}_{\mathcal{L}}$ with sampled values $\tau_{\mathcal{L}} \sim \mathrm{T}_{\mathcal{L}}$ and expected value $\bar{\tau}_{\mathcal{L}} := \mathbb{E}[\mathrm{T}_{\mathcal{L}}]$. $N_{\mathcal{L}}^*$ will denote the number of learning processes such that all $N_{\mathcal{L}} \geq N_{\mathcal{L}}^*$ include at least one transition learned from for each environment transition.

**Question 4:** *How does $N_{\mathcal{L}}^*$ scale with $\bar{\tau}_{\mathcal{L}}$ and the number of parameters $|\theta|$?*

**Figure 8:** In Figure 8a we demonstrate that $N_{\mathcal{L}}^*$ grows approximately linearly with $\bar{\tau}_{\mathcal{L}}$. This scaling is in line with what we would expect for the round-robin algorithm with large networks [50]. Additionally, our results in Figure 8b appear to also showcase linear scaling of $N_{\mathcal{L}}^* \geq 1$ with $|\theta|$. We plot three different batch sizes per learning process in each figure to highlight the tradeoff between efficiency in achieving a particular amortized learning throughput and the number of changes made to

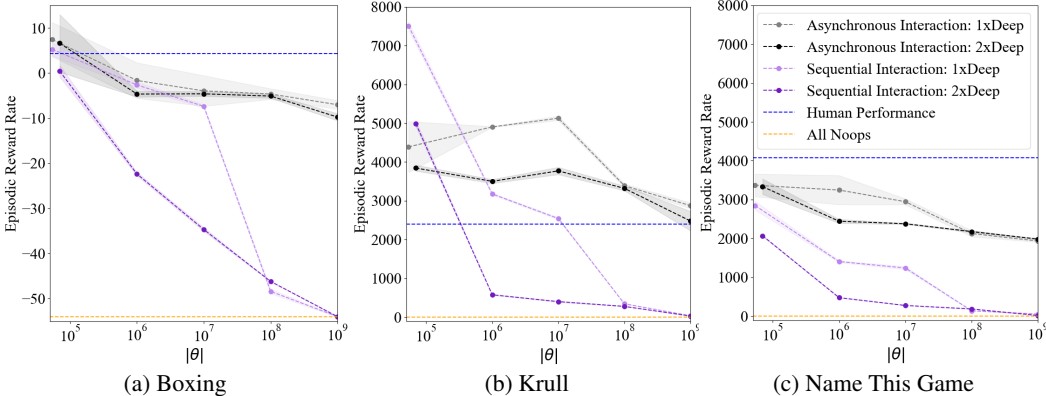

(a) Boxing       (b) Krull       (c) Name This Game

Figure 6: **Realtime Atari Performance vs. $|\theta|$.** The average episodic return over 2,000 simulated learning episodes. We compare models with a single inference process to those that perform staggered asynchronous inference following Algorithm 1 in **a)** Boxing, **b)** Krull, and **c)** Name This Game (see Figure 10). Human performance was reported by [61]. Small confidence intervals may be hard to see.

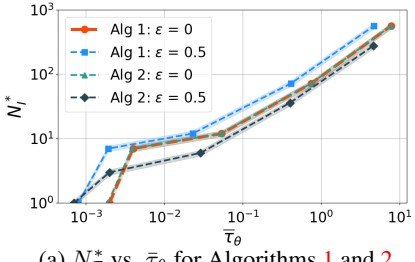 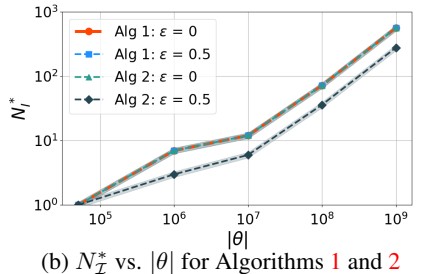

(a) $N_{\mathcal{I}}^*$ vs. $\bar{\tau}_\theta$ for Algorithms 1 and 2      (b) $N_{\mathcal{I}}^*$ vs. $|\theta|$ for Algorithms 1 and 2

Figure 7: **a)** We plot the scaling behavior of the inference compute requirement $N_{\mathcal{I}}^*$ as the expected action inference time $\bar{\tau}_\theta$ increases for ResNet polices across CPUs in the Game Boy environment. **b)** We plot the scaling behavior of $N_{\mathcal{I}}^*$ instead as a function of the model size $|\theta|$.

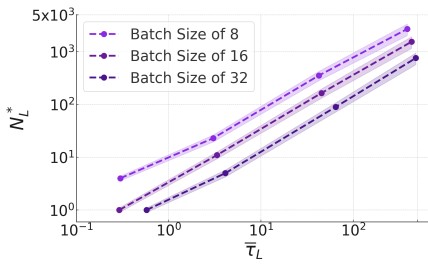 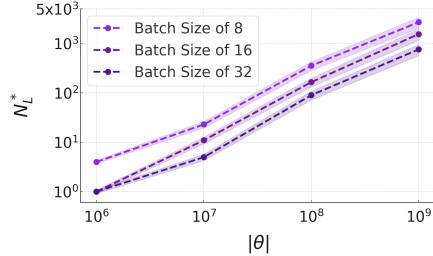

(a) $N_{\mathcal{L}}^*$ vs. $\bar{\tau}_{\mathcal{L}}$ for Round-Robin Learning      (b) $N_{\mathcal{L}}^*$ vs. $|\theta|$ for Round-Robin Learning

Figure 8: **a)** We plot the scaling behavior of the learning compute requirement $N_{\mathcal{L}}^* \geq 1$ as the expected transition learning time $\bar{\tau}_{\mathcal{L}}$ increases for ResNet polices across CPUs in the Game Boy environment. **b)** We plot the scaling behavior of $N_{\mathcal{L}}^*$ instead as a function of the model size $|\theta|$.

the parameters in the inference process. Higher batch sizes result in longer and increasingly parallel updates with better amortized learning throughput using the same number of processes and fewer changes to the parameters used for inference (as highlighted by $\bar{\tau}_{\mathcal{L}}$). Meanwhile, smaller batch size result in shorter and increasingly sequential updates that lead to worse amortized learning throughput using the same number of processes and more changes to the parameters used for inference.

## 6 DISCUSSION

**Implications for Software:** One difficulty in deploying RL environments within realtime settings is limitations in current APIs such as Open AI gym [10]. In realtime settings, the environment needs to run in its own separate process so that it is not blocked by the agent (just like the real world). We hope that the contribution of our code and environments can thus help jump-start the community's ability to conduct research on this important setting. A new paradigm of agent-environment interaction is needed where 1 process is used for the environment with $N_{\mathcal{I}}$ processes dedicated for inference and $N_{\mathcal{L}}$ processes dedicated for learning, depending on specific resource constraints.

**Implications for Hardware:** Our paper demonstrates the benefits of asynchronous computation within realtime settings and thus advances in hardware that enable this will serve to amplify the impact of our findings. Memory bandwidth is a primary bottleneck in allowing for asynchronous computation with current hardware. So any improvements i.e. in GPU memory bandwidth, in bandwidth across nodes, or the number of GPUs or CPUs per node will make the scalability of asynchronous approaches increasingly viable. Taking a longer-term perspective, hardware architectures that move beyond the von Neumann seperation of memory and compute, such as so called "neuromorphic" computing, will also serve to enable larger scale asynchronous computation like we see in the brain.

**Bigger Models:** In this paper, we have taken a deeper look at RL in realtime settings and the viability of increasing the neural network model size in these environments. Our theoretical analysis of regret bounds has demonstrated the downfall of models that implement a single action inference process as model sizes grow (Remark 1) and we have proposed staggering algorithms that address this limitation for environments that are sufficiently deterministic (Remark 2). Our empirical results playing realtime games corroborate these findings and demonstrate the ability to perform well with models that are orders of magnitude larger. While conventional wisdom often leads researchers to think that smaller models are necessary for realtime settings, our work indicates that this is not necessarily the case and takes a step towards making realtime foundation model deployment realistic.

## Reproducibility Statement

In this paper, we fully disclose all the necessary information to reproduce the main experimental results, as detailed in both Section 5 and Appendix A. We have also released all the necessary code along with detailed instructions to reproduce our results at https://github.com/CERC-AAI/realtime_rl. Experimental settings, including hyperparameters needed to reproduce our results are clearly outlined in the paper and further elaborated on in the appendix. Moreover, we provide appropriate statistical significance measures in our plots when applicable, such as shading in Figures 4 and 5 to communicate run-to-run variance in terms of 95% confidence intervals. We also include comprehensive information on the compute resources, such as the type of hardware, memory, and time required for execution, as outlined in the "Implementation Details" paragraph of Section 5. This ensures that our results are reproducible, and necessary computational resources are well-specified for accurate replication.

## Acknowledgements

We would like to acknowledge our support from the Canada CIFAR AI Chair Program and from the Canada Excellence Research Chairs (CERC) Program. We also thank the IBM Cognitive Compute Cluster, the Mila cluster, and Compute Canada for providing computational resources. We are grateful to Yoshua Bengio for his support during initial discussions that helped shape the direction of this paper. Moreover, we would like to thank Johan Obando-Ceron, Khurram Javed, Mark Ring, and Rich Sutton for valuable feedback that helped shape the framing of our contribution. We also thank Olexa Bilaniuk for his assistance with questions related to computational resources and multiprocessing.

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

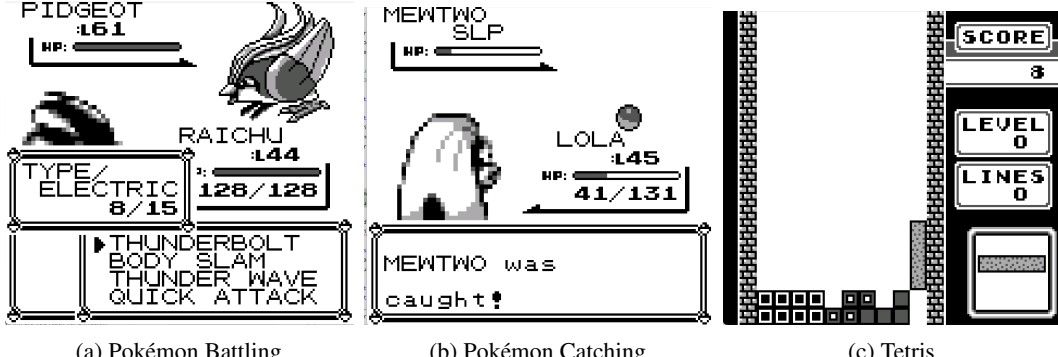

| (a) Pokémon Battling | (b) Pokémon Catching | (c) Tetris |

Figure 9: **a)** A frame from the final battle of Pokémon Blue when the agent is deciding on the next move. **b)** A frame from the final catching encounter of Pokémon Blue when the agent has just successfully caught Mewtwo. **C)** A frame from Tetris right before the agent completes its first line.

[106] Yufeng Yuan and A Rupam Mahmood. Asynchronous reinforcement learning for real-time control of physical robots. In *2022 International Conference on Robotics and Automation (ICRA)*, pp. 5546–5552. IEEE, 2022.

[107] Yunzhi Zhang, Ignasi Clavera, Boren Tsai, and Pieter Abbeel. Asynchronous methods for model-based reinforcement learning. *arXiv preprint arXiv:1910.12453*, 2019.

[108] Modjtaba Shokrian Zini, Mohammad Pedramfar, Matthew Riemer, Ahmadreza Moradipari, and Miao Liu. Coagent networks revisited. *arXiv preprint arXiv:2001.10474*, 2020.

## A    FURTHER DETAILS SUPPORTING THE MAIN TEXT

**Software Libraries:** Our experiments leverage Numpy [24], which is publicly available following a BSD license. Neural network models were developed using Pytorch [70], which is publicly available following a modified BSD license. The Gym Retro project [65] used to simulate the Game Boy in a RL environment is made available following a MIT license. We are not at liberty to distribute the proprietary ROMs associated with Pokémon or Tetris and each person that deploys our provided code must separately obtain their own copy.

**Environment Details:** We depict the environments considered in our paper in Figure 9. We consider six discrete actions for both Pokémon Blue and Tetris including the A button, the B button, the left directional button, the up directional button, the right directional button, and the down directional button. In the Battling Environment when the opponent Pokémon is knocked out by the agent's Pokémon a reward of 1 is received and a reward of $-1$ is received when a users Pokémon is knocked out. Battles include 1-6 Pokémon for the agent and 1-6 Pokémon for the opponent AI. In the Catching Environment a reward of 1 is received by the agent when a wild Pokémon is captured and $-1$ when the encounter is terminated unsuccessfully. In Tetris we provide changes in the in-game score as a reward for the agent to learn from. We leverage Atari environments as provided by Gymnasium [101], specifically using the "NoFrameSkip-v4" variants with action 0 taken for "noops" to simulate real-world play on a console.

**Training Procedure for Tetris:** The one episode of human play provided for Tetris included 16,106 steps where non-noop actions were taken. We used our experiments from Figure 7 to calculate the amount of action delay per step for each model and populated the replay buffer with 16,106 transitions corresponding to these actions with observations delayed by the expected amount for each model. We the trained each model for 16,000 steps before tuning the model in a simulation of the environment with the corresponding amount of delay and inaction for 2,000 episodes with one update on the replay buffer for each action taken in the environment. The episodic reward from Figure 5 corresponds to the average episodic reward achieved during that training period. To ensure differences in learning are only the results of delay and inaction while ignoring differences based on different learning speeds resulting from the model size itself, we use the delay and inaction calculated for models of that size, but then always learn a model of the standard 15 convolutional layer ResNet depth with $k = 1$ filter

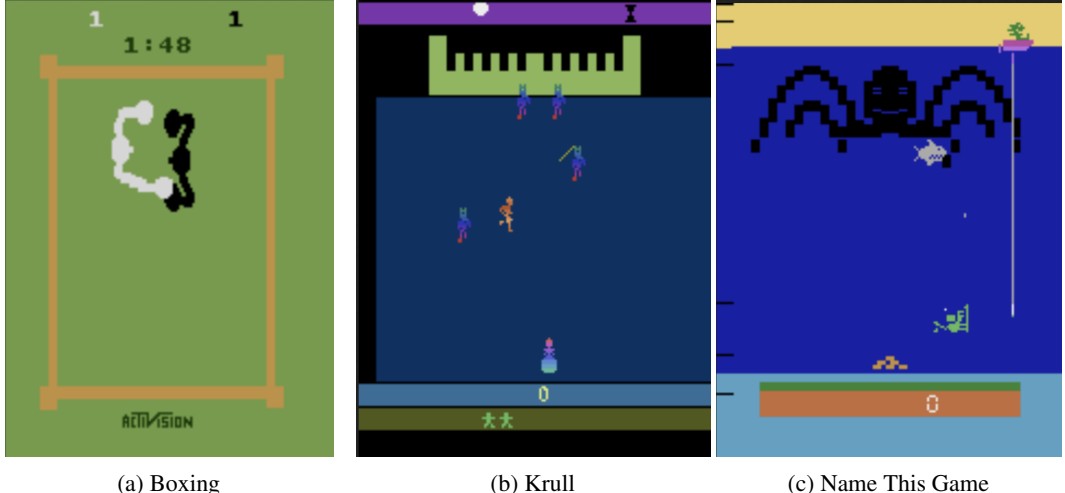

(a) Boxing          (b) Krull          (c) Name This Game

Figure 10: **Realtime Atari Environments.** Frames from the Atari environments **a)** Boxing, **b)** Krull, and **c)** Name This Game. These particular games were selected because DQN learning is known to achieve comparable performance to humans within 2,000 episodes when trained without delay.

size. We also tried using the same model size, which followed the same general trend, but report it this way to address a concern mentioned by a reviewer of this paper.

**Training Procedure for Atari:** We followed a similar training procedure for Atari. We used our experiments from Figure 7 to calculate the amount of action delay per step for each model. The number of delay and inaction steps were normalized for the slightly faster 60 frame per second environment. This only has an impact for the 1B model size, which now has a delay of 599 steps rather than 596 steps for the standard depth ResNet model. The episodic reward from Figure 6 corresponds to the average episodic reward achieved during that training period. For the 2xDeep architecture we leveraged a ResNet model with 6 blocks rather than 3 blocks, which corresponds to 30 convolutional layers rather than 15 convolutional layers. In the final three blocks, each layers has $32k$ filters where $k = 0.0625$ corresponds to 70k parameters, $k = 0.75$ corresponds to 1M parameters, $k = 4.75$ corresponds to 10M parameters, $k = 18.5$ corresponds to 100M parameters, and $k = 63$ corresponds to 1B parameters. As is standard practice for Atari, we now learn every 4 environment steps. However, we do not skip frames and allow models to act at every step if it can.

**Statistical Significance:** We also note that error bar shading throughout our paper reflects 95% confidence intervals computed with three random seeds: 0, 1, and 2.

**Limitations:** In both our experiments on Pokémon Blue and Tetris, performance is well below human-level. This is because both of these games pose significant exploration problems and we train our models from scratch for a limited amount of time. We believe that these experiments are more than sufficient to showcase the benefits of staggered asynchronous inference in comparison to the sequential interaction framework by showcasing when the latter framework breaks down in realtime settings. However, we speculate that the results showing that game play does not suffer despite significant action delay will likely not generalize to more intricate human-level policies. We also corroborate our findings in a setting where agents can perform on a human-level with limited training episodes for the Atari games Boxing, Krull, and Name This Game.

**Algorithm Pseudocode:** Here we provide detailed pseudocode for Algorithm 2, which could not be included in the main text due to space constraints.

A.1    POKÉMON RESULTS WITH SMALLER MODEL SIZES

Due to space restrictions we were also not able to include our experiments for the Pokémon battling and catching domains with $|\theta| < 100M$ in the main text. We provide these results for $|\theta| = 1M$ in Figure 11 and $|\theta| = 10M$ in Figure 12. As expected by our theory, the difference between asynchronous interaction and sequential interaction is expected to be less when the action inference

---

**Algorithm 2** Expected Time Inference Staggering

---

**Initialize:** $\hat{\bar{\tau}}_\theta = 0$, $\tau_{\text{tot}} = 0$, $a_{\text{tot}} = 0$, and delay[processnum] $= 0 \ \forall$ processnum $\in [1, ..., N_\mathcal{I}]$
**Run:** INFERENCE[processnum] $\forall$ processnum $\in [1, ..., N_\mathcal{I}]$
  **function** INFERENCE(processnum)
    **while** alive **do**
      sleep(delay[processnum])         ▷ Sleep for any delays accumulated by other processes
      delay[processnum] $\leftarrow 0$
      $a, \tau_\theta \sim \pi_\theta(s_t)$         ▷ Have the policy sample an action and inference time
      $a_{\text{tot}} \leftarrow a_{\text{tot}} + 1$
      $\tau_{\text{tot}} \leftarrow \tau_{\text{tot}} + \tau_\theta$
      $\hat{\bar{\tau}}'_\theta \leftarrow \tau_{\text{tot}} / a_{\text{tot}}$
      $\delta\tau \leftarrow \hat{\bar{\tau}}'_\theta - \hat{\bar{\tau}}_\theta$
      **if** $\delta\tau \geq 0$ **then**         ▷ Wait more further from the current process
        **for** num $\neq$ processnum $\in [1, ..., N_\mathcal{I}]$ **do**
          delay[num] $\leftarrow$ delay[num] + dist(num, processnum) abs($\delta\tau$)$/N_\mathcal{I}$
      **else**         ▷ Wait more closer to the current process
        **for** num $\neq$ processnum $\in [1, ..., N_\mathcal{I}]$ **do**
          delay[num] $\leftarrow$ delay[num] + $(N_\mathcal{I} - 1)$dist(num,processnum) abs($\delta\tau$)$/N_\mathcal{I}$
      $a_{t+\lceil \tau_\theta / \tau_\mathcal{M} \rceil} \leftarrow a$         ▷ Register action in environment

---

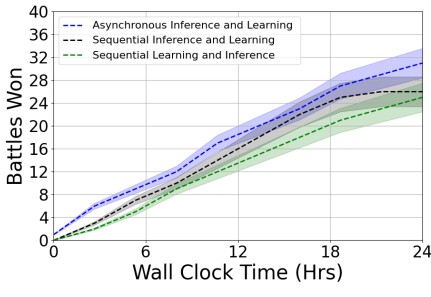
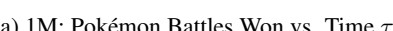

(a) 1M: Pokémon Battles Won vs. Time $\tau$

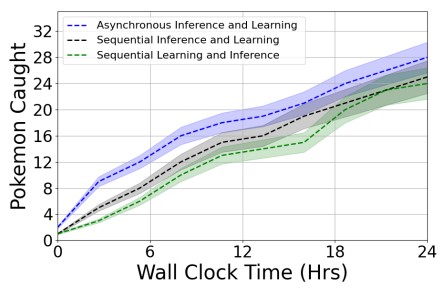

(b) 1M: Wild Pokémon Caught vs. Time $\tau$

Figure 11: **Realtime Pokémon Performance for Staggered Asynchronous Interaction & Learning.**
**a)** Battles won in Pokémon Blue as a function of time for $|\theta| = 1M$. **b)** Wild Pokémon caught in Pokémon Blue as a function of time for $|\theta| = 1M$.

time is low. In this experiment, the sequential interaction baseline acts every 5 steps at 1M parameters, 8 steps at 10M parameters, and 70 steps at 100M parameters. For the Pokemon game it is expected that there is not much improvement acting at every step as it is well known to speed runners that in common circumstances the game could take as many as 17 frames to respond to non-noop actions in which time intermediate actions are "buffered" and not yet registered in the environment [92]. This is a fact commonly exploited by speed-runners to allow for them to manipulate the RNG of the game and execute actions in a "frame perfect" manner despite natural human imprecision when it comes to action timing. As a result, the overall performance in this domain is in line with expectations as differences are most significant when inference times are greater than this 17 environment step threshold. The results reported in Figure 11 and in Figure 12 are mean performances along with 95% confidence intervals, calculated from 10 runs with different random seeds.

## A.2 PERFORMANCE AS A FUNCTION OF NON-NOOP ACTIONS

In Figures 13a and 13b we observe that the difference between progress through the game for sequential and asynchronous algorithms per non-noop action is not statistically significant in the Pokémon battling and catching environments. As such, our asynchronous algorithm takes full advantage of the increased throughput of non-noop actions taken in the environment.

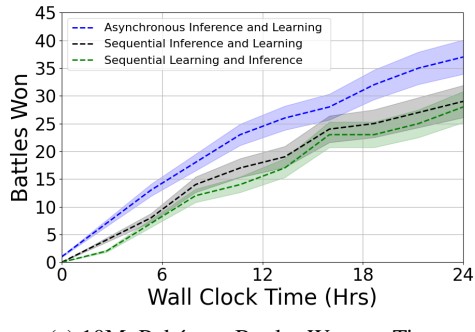
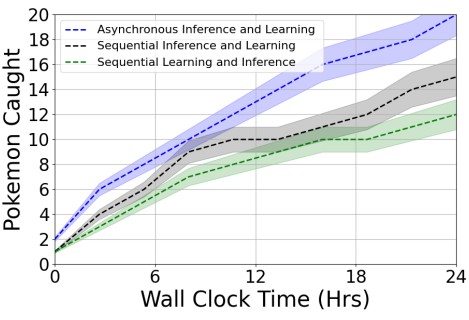

(a) 10M: Pokémon Battles Won vs. Time $\tau$  (b) 10M: Wild Pokémon Caught vs. Time $\tau$

Figure 12: **Realtime Pokémon Performance for Staggered Asynchronous Interaction & Learning.**
**a)** Battles won in Pokémon Blue as a function of time for $|\theta| = 10M$. **b)** Wild Pokémon caught in Pokémon Blue as a function of time for $|\theta| = 10M$.

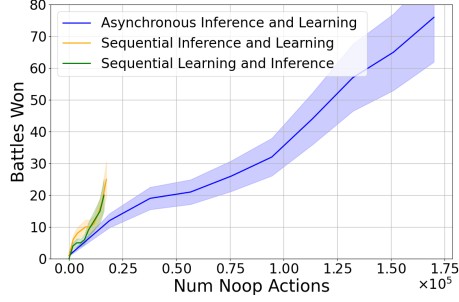
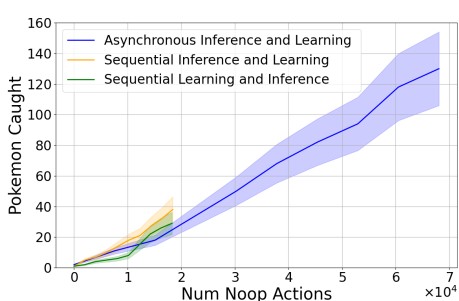

(a) 100M: Pokémon Battles Won vs. Number of Non-Noop Actions  (b) 100M: Wild Pokémon Caught vs. Number of Non-Noop Actions

Figure 13: **Realtime Pokémon Performance for Staggered Asynchronous Interaction & Learning.**
**a)** Battles won in Pokémon Blue as a function of non-noop actions taken in the environment for $|\theta| = 100M$. **b)** Wild Pokémon caught in Pokémon Blue as a function of non-noop actions taken in the environment for $|\theta| = 100M$.

## A.3 RAINBOW EXPERIMENTS

We now also include results with Rainbow [26] used for learning rather than the DQN on Atari. We again use a standard $k = 1$ filter size at the standard 15 convolutional layer ResNet depth now with the Rainbow model for learning based on model inference times that correspond to those used for the DQN. Note that Rainbow performing worse than the DQN in the Boxing environment, even without delay, was reported in Table 5 of the original paper [26]. Our implementation of Rainbow draws heavily from the following repository https://github.com/davide97l/Rainbow/tree/master with hyperparameters taken from that repository other than the 0.0001 learning rate corresponding to our other Atari experiments. These experiments yet again showcase the superior ability of staggered asynchronous inference to maintain performance at larger model sizes and inference times.

## B PROOFS FOR EACH THEORETICAL STATEMENT

Our proofs rely on the following core assumptions, restated from the main text:

1. The environment step time can be treated as an independent random variable $T_\mathcal{M}$ with sampled values $\tau_\mathcal{M} \sim T_\mathcal{M}$ and expected value $\bar{\tau}_\mathcal{M} := \mathbb{E}[T_\mathcal{M}]$.

2. The environment interaction time can be treated as an independent random variable $T_\mathcal{I}$ with sampled values $\tau_\mathcal{I} \sim T_\mathcal{I}$ and expected value $\bar{\tau}_\mathcal{I} := \mathbb{E}[T_\mathcal{I}]$.

3. The action inference time of the policy can be treated as an independent random variable $T_\theta$ with sampled values $\tau_\theta \sim T_\theta$ and expected value $\bar{\tau}_\theta := \mathbb{E}[T_\theta]$.

4. Asynchronous learning can learn from every interaction with $\tilde{\mathcal{M}}_{\text{delay}}$.

Issues with the proof in an earlier workshop version of the paper [85] have been corrected here.

## B.1 DEFINITION 1

Most of Definition 1 just recaps the dynamics of how the agent interacts with an asynchronous ground MDP following assumptions 1-3 about the nature of that interaction. All that is left to show is that this can be viewed as a delayed MDP and that it can be viewed as a semi-MDP. The interaction process highlighted in Definition 1 matches that of a Random Delay Markov Decision Process (RDMDP) [9] where the action delay distribution is defined by the random variable $\lceil \tau_\theta / \tau_\mathcal{M} \rceil$. To show it is a semi-MDP as well, we consider the same proof style of Theorem 1 in Sutton et al. [95]:

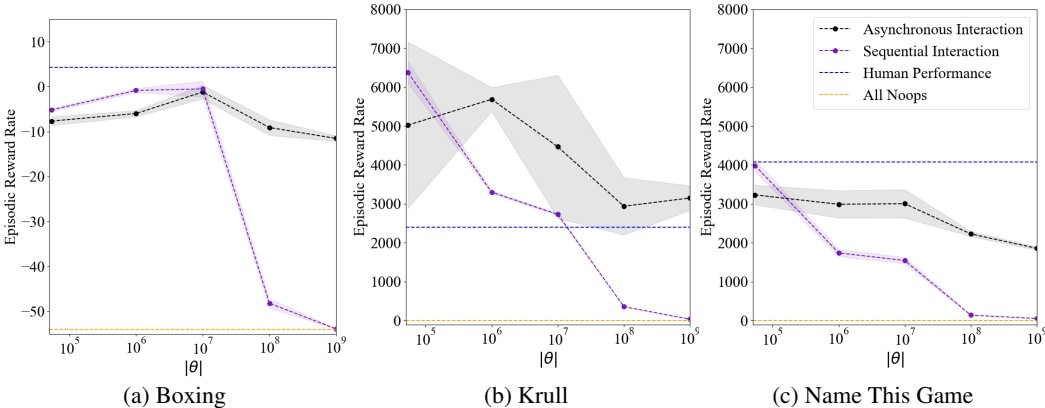

(a) Boxing      (b) Krull      (c) Name This Game

Figure 14: **Realtime Atari Rainbow Performance vs.** $|\theta|$. The average episodic return over 2,000 simulated learning episodes. We compare Rainbow models with a single inference process to staggered asynchronous inference following Algorithm 1 in **a)** Boxing, **b)** Krull, and **c)** Name This Game (see Figure 10). Human performance was originally reported by [61]. Small confidence intervals may be hard to see.

A semi-MDP consists of (1) a set of states, (2) a set of actions, (3) for each pair of state and action, an expected cumulative discounted reward, and (4) a well-defined joint distribution of the next state and transit time. We now demonstrate each of these properties. The set of states is $\mathcal{S}$ and the set of actions is $\mathcal{A}$. The expected reward and the next-state and transit-time distributions are well defined for every state and delayed action. These expectations and distributions are well defined because $\mathcal{M}_{\text{async}}$ is Markov, thus the next state, reward, and time are dependent only on the delayed action chosen and the state in which it was initiated. Transit times with arbitrary real intervals are permitted in semi-MDPs.

### B.2 THEOREM 1

To prove Theorem 1 we will demonstrate the validity of each equation of the theorem following the order of presentation in the main text.

**Equation 1:** By definition $\Delta_{\text{learn}}(\tau)$ and $\Delta_{\text{inaction}}(\tau)$ must be independent contribution to the total regret because learning regret is only incurred when acting in the environment following $\pi$ and inaction regret is only incurred when not acting in the environment and thus following the default behavior policy $\beta$. The interaction frequency does not depend on the parameter values of $\pi$ as they change following from the independent random variable assumption. Even when regret from learning is eliminated and regret from inaction is eliminated there is still another independent source of regret that persists $\Delta_{\text{delay}}(\tau)$ reflecting the lower reward rate of the best possible policy in acting over $\tilde{\mathcal{M}}_{\text{delay}}$ in comparison to the best possible policy acting over $\mathcal{M}_{\text{async}}$.

**Equation 2:** The worst case lower bound for the standard notion of regret arising from the need for learning and exploration has been established as $\Delta_{\text{learn}}(\mathcal{T}) \in \Omega(\sqrt{\mathcal{T}})$ where $\mathcal{T}$ denotes the number of discrete learning steps taken in the environment. Given that we learn from every asynchronous interaction with the environment following assumption 4, then the regret as a function of $\tau$ scales with the expected number of discrete environment steps as a function of $\mathcal{T}$ i.e. $\mathbb{E}[\mathcal{T}(\tau)] = \tau/\bar{\tau}_{\mathcal{I}}$ because $T_{\mathcal{M}}$ and $T_{\mathcal{I}}$ are independent. Therefor, $\Delta_{\text{learn}}(\tau) \in \Omega(\sqrt{\tau/\bar{\tau}_{\mathcal{I}}})$. As noted in the footnote in the main text, this analysis equally applies to the known $\Delta_{\text{learn}}(\mathcal{T}) \in \tilde{\mathcal{O}}(\sqrt{\mathcal{T}})$ minimum upper bound [68].

**Equation 3:** The worst case lower bound on $\Delta_{\text{inaction}}(\tau)$ is derived by considering a worst case environment with two actions $a_1$ and $a_2$ and two states $s_1$ and $s_2$ where the default behavior $\beta$ takes its own action $a_3$ at every state. The reward provided is 1 at $s_1$ and 0 at $s_2$. The next state is $s_1$ regardless of the state if either $a_1$ or $a_2$ is taken and $s_2$ if $a_3$ is taken. In this environment the optimal reward rate is 1 and the reward rate when following $\beta$ is 0.0. The expected number of times $a_3$ is taken during $\tau$ seconds in $\mathcal{M}_{\text{async}}$ is then $\tau/\bar{\tau}_{\mathcal{I}} \times (\bar{\tau}_{\mathcal{I}} - \bar{\tau}_{\mathcal{M}})/\bar{\tau}_{\mathcal{M}}$ because $\beta$ is used for $(\bar{\tau}_{\mathcal{I}} - \bar{\tau}_{\mathcal{M}})/\bar{\tau}_{\mathcal{I}}$ percent of ground MDP actions and the expected number of actions taken over $\tau$ is $\tau/\bar{\tau}_{\mathcal{M}}$ when $T_{\mathcal{M}}$ and $T_{\mathcal{I}}$ are independent. Therefor, $\Delta_{\text{inaction}}(\tau) \in \Omega((\tau/\bar{\tau}_{\mathcal{I}}) \times (\bar{\tau}_{\mathcal{I}} - \bar{\tau}_{\mathcal{M}})/\bar{\tau}_{\mathcal{M}})$ for this particular environment. Meanwhile, the expected inaction regret is also upper bounded by $\Delta_{\text{inaction}}(\tau) \leq r_{\max} \times (\tau/\bar{\tau}_{\mathcal{I}}) \times (\bar{\tau}_{\mathcal{I}} - \bar{\tau}_{\mathcal{M}})/\bar{\tau}_{\mathcal{M}}$ where $r_{\max}$ is the maximum possible reward per step because by definition the agent cannot incur regret from inaction when it is acting in the environment. Therefor, we have demonstrated that Equation 3 holds.

**Equation 4:** The worst case lower bound on $\Delta_{\text{delay}}(\tau)$ is derived by considering a worst case environment with $n$ states $\mathcal{S} = \{s_1, ..., s_n\}$ and $n$ actions $\mathcal{A} = \{a_1, ..., a_n\}$ where the default behavior $\beta$ takes its own action $a_0$ at every state. If the agent takes the action $a_i$ corresponding to the state $s_i$ for any $i \in \{1, ..., n\}$ the agent receives a reward of 1, otherwise it receives a reward of 0. The agent stays in the current state $s_i$ regardless of the action with probability $p_{\text{minimax}}$ and goes to the next state in the cycle $s_{i+1}$ with probability $1 - p_{\text{minimax}}$ where $p_{\text{minimax}} \geq 1/n$ because the sum over next state probabilities must equal 1. After state $s_n$ the agent returns to state $s_1$. The probability of staying in the current state for $k$ consecutive environment steps is $(p_{\text{minimax}})^k$ and in the limit as $n \to \infty$ this is also the probability that the agent is in the current state $k$ steps later. If the agent is not in the current state $k$ steps later, the agent will apply a sub-optimal action based on the old state because no state will overtake the current state as more likely than the current state with the state likelihoods converging to uniform in the limit as $k \to \infty$ when the Markov chain fully mixes. So the regret per ground environment step incurred by the optimal policy that acts with $k$ step lag is $1 - (p_{\text{minimax}})^k$. Thus, the best expected regret rate per step that can be ensured with actions delayed by $k = \lceil \tau_\theta/\tau_{\mathcal{M}} \rceil$ is $\geq \mathbb{E}[1 - (p_{\text{minimax}})^{\lceil \tau_\theta/\tau_{\mathcal{M}} \rceil}]$. Moreover, the expected number of steps

taken in the asynchronous MDP over $\tau$ seconds is $\tau/\bar{\tau}_{\mathcal{I}}$, so we can lower bound the expected regret as $\Delta_{\text{delay}}(\tau) \geq (\tau/\bar{\tau}_{\mathcal{I}}) \times \mathbb{E}[1 - (p_{\text{minimax}})^{\lceil \tau_\theta/\tau_{\mathcal{M}} \rceil}]$. This then leads us to the conclusion of Equation 4 from the main text that $\Delta_{\text{delay}}(\tau) \in \Omega((\tau/\bar{\tau}_{\mathcal{I}}) \times \mathbb{E}[1 - (p_{\text{minimax}})^{\lceil \tau_\theta/\tau_{\mathcal{M}} \rceil}])$.

**Tighter Version of Equation 4:** While the definition of $p_{\text{minimax}}$ in the main text holds for the counter example above, this example relies on the fact that transitioning states actually has an impact on the reward of a policy and its value function. In general, it does not matter if the state changes if the change does not impact the optimal policy. So a tighter version of $p_{\text{minimax}}$ would be defined over a $\pi^*$-irrelevance state abstraction $\phi_{\pi^*}$ of the state space rather than the ground state space, following the terminology of [52]. In a $\phi_{\pi^*}$ abstraction every abstract state in $\phi_{\pi^*}(\mathcal{S})$ has an action $a^*$ that is optimal for all the ground states $i$ and $j$ in that abstract state. As a result, $\phi_{\pi^*}(s_i) = \phi_{\pi^*}(s_j)$ implies that $\max_{a \in \mathcal{A}} Q^{\pi^*}(s_i, a) = \max_{a \in \mathcal{A}} Q^{\pi^*}(s_j, a)$. For example, a tighter version of Equation 4 could be written with $p_{\text{minimax}} := \min_{s \in \mathcal{S}, a \in \mathcal{A}} \max_{\phi_{\pi^*}(s_i) \in \phi_{\pi^*}(\mathcal{S})} \sum_{\phi_{\pi^*}(s_j) = \phi_{\pi^*}(s_i)} p(s_j|s, a)$.

## B.3 REMARK 1

We restate the derivation of the remark from the main text, filling in a bit more detail for clarity. When $\pi$ and $\mathcal{M}_{\text{async}}$ interact sequentially, we must have $\tau_{\mathcal{I}} \geq \tau_\theta$, so $\Delta_{\text{inaction}}(\tau) \in \Omega(\tau/\bar{\tau}_{\mathcal{I}} \times (\bar{\tau}_{\mathcal{I}} - \bar{\tau}_{\mathcal{M}})/\bar{\tau}_{\mathcal{M}}) \in \Omega(\tau/\bar{\tau}_\theta \times (\bar{\tau}_\theta - \bar{\tau}_{\mathcal{M}})/\bar{\tau}_{\mathcal{M}})$. This implies that even as $\tau \to \infty$, the worst case regret rate $\Delta_{\text{realtime}}(\tau)/\tau \in \Omega(\Delta_{\text{inaction}}(\tau)/\tau) \in \Omega((\bar{\tau}_\theta - \bar{\tau}_{\mathcal{M}})/\bar{\tau}_{\mathcal{M}}\bar{\tau}_\theta)$ following from Theorem 1.

## B.4 REMARK 2

**Interaction Time of Algorithm 1:** At any point in time, by definition $\hat{\tau}_\theta^{\text{max}} \leq \tau_\theta^{\text{max}}$ as the estimated maximum must be less than or equal the current maximum. Therefore, with $N_{\mathcal{I}}$ equally spaced processes, the interaction time must be $\tau_{\mathcal{I}} = \lceil \hat{\tau}_\theta^{\text{max}}/N_{\mathcal{I}}\tau_{\mathcal{M}} \rceil \times \tau_{\mathcal{M}}$ and Algorithm 1 ensures equal spacing between processes whenever $\hat{\tau}_\theta^{\text{max}}$ does not change. Regardless of the changing value of $\hat{\tau}_\theta^{\text{max}}$, we can also bound the individual interaction time steps $\tau_{\mathcal{I}} \leq \lceil \tau_\theta^{\text{max}}/N_{\mathcal{I}}\tau_{\mathcal{M}} \rceil \times \tau_{\mathcal{M}}$ and correspondingly the global average $\bar{\tau}_{\mathcal{I}} \leq \tau_\theta^{\text{max}}/N_{\mathcal{I}}$ when $\bar{\tau}_{\mathcal{I}} > \bar{\tau}_{\mathcal{M}}$ and $\bar{\tau}_{\mathcal{I}} = \bar{\tau}_{\mathcal{M}}$ otherwise.

**Interaction Time of Algorithm 2:** As $\tau \to \infty$, $\hat{\tau}_\theta \to \bar{\tau}_\theta$ due to the law of large numbers and thus $\delta\tau \to 0$, which implies that additional waiting times become zero in the limit. Therefore, with $N_{\mathcal{I}}$ equally spaced processes, the interaction time must be $\tau_{\mathcal{I}} = \lceil \bar{\tau}_\theta/N_{\mathcal{I}}\tau_{\mathcal{M}} \rceil \times \tau_{\mathcal{M}}$ and Algorithm 2 ensures equal spacing between processes whenever $\hat{\tau}_\theta$ does not change and $\delta\tau = 0$. Thus, as $\tau \to \infty$ the global average is $\bar{\tau}_{\mathcal{I}} = \bar{\tau}_\theta/N_{\mathcal{I}}$ when $\bar{\tau}_{\mathcal{I}} > \bar{\tau}_{\mathcal{M}}$ and $\bar{\tau}_{\mathcal{I}} = \bar{\tau}_{\mathcal{M}}$ otherwise.

**Bringing it Together:** Algorithm 1 is capable of scaling the expected interaction time with the number of processes by $\bar{\tau}_{\mathcal{I}} \leq \min(\tau_\theta^{\text{max}}/N_{\mathcal{I}}, \bar{\tau}_{\mathcal{M}})$ where $\tau_\theta^{\text{max}}$ is the maximum encountered value of $\tau_\theta$ as $\tau \to \infty$. This then implies that for $N_{\mathcal{I}} \geq N_{\mathcal{I}}^* = \lceil \tau_\theta^{\text{max}}/\bar{\tau}_{\mathcal{M}} \rceil$, $\bar{\tau}_{\mathcal{I}} = \bar{\tau}_{\mathcal{M}}$. Algorithm 2 is capable of scaling the expected interaction time with the number of processes by $\bar{\tau}_{\mathcal{I}} = \min(\bar{\tau}_\theta/N_{\mathcal{I}}, \bar{\tau}_{\mathcal{M}})$ as $\tau \to \infty$ following the law of large numbers. This then correspondingly implies that for $N_{\mathcal{I}} \geq N_{\mathcal{I}}^* = \lceil \bar{\tau}_\theta/\bar{\tau}_{\mathcal{M}} \rceil$, $\bar{\tau}_{\mathcal{I}} = \bar{\tau}_{\mathcal{M}}$.

## B.5 COMPARISON WITH ACTION CHUNKING APPROACHES

The *action chunking* approach learns a policy that produces multiple actions at a time with a single inference step i.e. a policy $\pi_\theta(a_t, ..., a_{t+k}|s_t)$ that produces $k$ actions at a time. Clearly action chunking does not address the root cause of regret from delay $\Delta_{\text{delay}}(\tau)$ that our asynchronous inference framework also suffers from as $k$ steps of delay is built directly into the policy. However, it is less clear on the surface how the action chunking approach relates to $\Delta_{\text{learn}}(\tau)$ and $\Delta_{\text{inaction}}(\tau)$.

**Does action chunking eliminate $\Delta_{\text{inaction}}(\tau)$?** Action chunking could eliminate regret from inaction if $\tau_\theta^{\text{max}}/k = \tau_{\mathcal{M}}$. But this is not possible for any value of $\tau_\theta^{\text{max}}$ with sufficiently large $k$ because $\tau_\theta^{\text{max}}$ must depend on $k$ as the size of the outputs produced scales linearly with $k$. So action chunking can eliminate inaction for some policy classes, but not in general as policy inference times become large. Our asynchronous inference approach provides a more general result in this regard (see Remark 2).

**Does action chunking impact $\Delta_{\text{learn}}(\tau)$?** A hidden term excluded from our bound of $\Delta_{\text{learn}}(\tau)$ in the main text is a dependence on the size of the action space $|\mathcal{A}|$ such that $\Delta_{\text{learn}}(\tau) \in \Omega(\sqrt{\tau|\mathcal{A}|}/\bar{\tau}_{\mathcal{I}})$ [31]. We suppressed this dependence for clarity in the main text, but it is important to keep in mind

when considering action chunking approaches as this implies that the regret lower bound scales with $\sqrt{|\mathcal{A}|^k}$ and thus can lead to an exponentially worse regret from learning $\Delta_{\text{learn}}(\tau)$ even in cases where $\Delta_{\text{inaction}}(\tau)$ is eliminated with sufficiently large $k$.

### B.6 Achieving Sublinear Regret with Staggered Asynchronous Inference

As we have demonstrated in Remark 2, when enough asynchronous threads are provided, a model of any size can eliminate regret from inaction. Regret from delay is not addressed by this approach, but goes to zero in deterministic environments. As a result, to demonstrate that total regret is sublinear for deterministic environments following Equation 1, we must only show that $\Delta_{\text{learn}}(\tau)$ can be upper bounded by a quantity sublinear in $\tau$. For example, if we use Q-Learning as we do in our experiments, there are known upper bounds for the case of utilizing optimistic exploration with a carefully designed learning rate of $\Delta_{\text{learn}}(T) \in \tilde{\mathcal{O}}(\sqrt{T})$ where $T$ is the number of steps in the environment [32]. In expectation $\mathbb{E}[T] = \tau/\bar{\tau}_{\mathcal{M}}$, so the upper bound on the expected regret will be $\Delta_{\text{learn}}(T) \in \tilde{\mathcal{O}}(\sqrt{\tau})$ as $\bar{\tau}_{\mathcal{M}}$ is a constant. As a result, our approach can achieve sublinear regret whenever $\Delta_{\text{delay}}(\tau)$ is sublinear, which includes, but is not limited to, all deterministic environments.

## C Towards Enabling Real-World Deployment of Agents in Even More Complex Scenarios

**Designing Safe Default Policies $\beta$:** In our work, we assume $\beta$ performs poorly such that it is unsafe and thus our goal with asynchronous interaction is to avoid using it altogether. However, an alternate research direction motivated by our formulation in Definition 1 would be to work on refining the $\beta$ function itself such that relying on it is guaranteed to be safe and not lead to irrevertible consequences. This is, for example, a major difference between the implementation of $\beta$ in the games Pokémon and Tetris. In Pokémon inaction does not prohibit the agent from eventually achieving success in the game, whereas in Tetris inaction leads to missed opportunities that can never be overcome. This materializes in our empirical results as we see a slower degradation of performance for the sequential interaction baseline as the model size grows for Pokémon. That said, the limitation of our asynchronous inference approach is that policies are based on previous states, which can itself be unsafe in stochastic environments. In this case, it would be logical to also consider beneficial and safe ways to modify $\beta$ itself, which can then just be viewed as another model that happens to have a faster inference time that should also be provided resources for asynchronous learning. This case closely mirrors the options framework [95; 4] and variants of it that consider deep hierarchies of policies [80; 82; 1]. As we do not want $\beta$ itself to lead the agent to OOD states, it probably would make sense to train this policy with a pessimism bias [49; 33] or a bias towards actions that make focused effects in the environment [3]. Moreover, our work is complimentary to efficient methods for performing alignment of large deep models [97], particularly for enhanced safety [98] and alignment with moral values [69; 18]. It is also worth noting that while neural scaling laws demonstrate generalization improvements with increased model size in the regime of having access to as much data as needed [34], capacity limits have often been found to improve generalization for RL [56; 59] and multi-agent RL [55; 58; 57]. As such, it may not make sense to scale model size indefinitely, as suggested by the general discourse of our paper, in data limited settings where safety is critical.

**Broader Domains with Diverse Subtasks or Nonstationarity:** When it comes to both learning a policy over the delayed semi-MDP and learning a default behavior policy, a key challenge is to adapt these policies to changing environment dynamics and new scenarios. Broadly, this is referred to as Continual RL (see [39] for a comprehensive survey). On the optimization end, these settings require consideration of the stability-plasticity dilemma. For example, we used recency based replay buffers in this work that have been standard in the RL literature since [61]. However, this recency based buffer maintenance strategy only makes sense when we are purely focused on plasticity. A reasonable alternative for continual learning would be buffers based on reservoir sampling [79] to emphasize stability or scalable memory efficient approximate buffers [77; 81; 6]. Another way to promote stability during learning is with regularization based approaches. For example, approaches that leverage knowledge distillation from old versions of models [53; 78] or approaches that penalize movement in parameter space [45]. In settings where subtasks are diverse it is also important to consider that the optimal interaction frequency may depend on the context. In this case, it may be beneficial to consider algorithms that adaptively select which layers to process at inference time

[86; 11; 12; 87], see [88] for a survey of approaches. More generally, this adaptive computation problem can be formalized within the coagent networks framework [99; 48; 108]. That said, the compositional generalization needed for utilizing these models in practice makes achieving real-world success very challenging [46]. Moreover, in the case of POMDPs, especially when they are large scale [83], it may make sense to adaptively change the interaction history context length sent to our policy to tradeoff the extent of modeling non-Markovian dependencies with inference and evaluation times [84].

**Interacting in Multi-agent Settings:** Another key consideration when generalizing our approach to more realistic settings is interacting with other agents in the environment. While asynchronous interaction can definitely help eliminate inaction, the amount of tolerable delay depends not just on the environment, but also on the behavior of other other agents in the environment. For example, the Boxing environment considered in our experiments from Figure 6 can also be played as a two agent game. We find that it is very difficult to perform on human-level in this environment even with delay as low as 3 environment steps, which ties back to the stochasticity of the policy used by the AI in single player mode. Another key challenge in multi-agent environments is when the policies of the other agents change i.e. due to learning, which makes the environment nonstationary from the perspective of each agent. When the agents are trained in a centralized fashion, they can be considered in each others updates to mitigate this nonstationarity [54]. However, in the more typical decentralized setting agents must build theory of mind models to speculate about the (changing) behavior of other agents [72] which must be carefully constructed for scalability in the presence of many other agents [60; 100]. Effective approaches have been developed to address this challenge by meta-learning with respect to the updating policies of other agents [22; 23; 42; 105; 43] or by learning to perform effective communication between agents [51; 67; 40; 41; 47]. In order to generalize our theoretical results in Theorem 1 to environments with multiple learning agents, we would have to consider regret with respect to complex game-theoretic solution concepts [44]. As such, we leave consideration of these approaches and exploration of a formulation for realtime regret in this setting to future work.

**Applicability Beyond RL:** In this paper, we focused on RL environments because of our use of the formalism of RL to establish our theoretical contributions. However, the general principles of this paper should apply to many practical domains that are not typically modeled using RL such as biomedical applications like peptide design [15], monitoring conversation topics across the internet [75; 25; 74; 38], and time series prediction based on incoming internet data [76]. Indeed, we chose to study the framework of RL as these problems can generally be considered a special case of the RL formalism [5]. For example, one broad application area of particular relevance to our paper is continual supervised learning. There are many potential settings of deployment related to this general topic that can all be considered special cases of the formulation we consider in this paper [66].

