# OpenReview forum: "Enabling Realtime Reinforcement Learning at Scale with Staggered Asynchronous Inference"
_ICLR.cc/2025/Conference — ICLR 2025 Poster_

### Official Review · Reviewer_VvDT · 2024-10-28

**Soundness:** 3
**Presentation:** 4
**Contribution:** 3
**Rating:** 8
**Confidence:** 3

**Summary:**

This paper highlights an often ignored discrepancy between the discrete-time RL framework and the real world. The world evolves while agents compute actions, limiting the types of problems agents can solve. Large models with high action inference and learning times conflict with intelligent agents' control over the environment. The paper examines this and explores asynchronous interaction paradigms for large models in high-frequency environments. It formalizes and tests the approach's benefits and limitations. Contributions include formalizing how time discretization induces a new learning problem, deriving regret bounds, proposing methods for staggered asynchronous inference, and conducting experiments on realtime games like Pokémon and Tetris.

**Strengths:**

1. Regret Decomposition seems to be rigorous in real-time reinforcement learning. This specific choice of time discretization is novel in Definition 1.
2. The staggered asynchronous learning and inference phase addresses  the poor scaling properties, highlighting that while the latter allows actions and learning at every step, actions are delayed and reflecting past states. It also emphasizes the importance of staggering processes to maintain regular intervals and shows implications for performance such as linear speedups.
3. Several Experiments clearly verify their theory.
4. The paper has great organization and is easy to follow.

**Weaknesses:**

* The experiment is recommended to be based on other base RL strategies rather than just DQN.
* Parallel updates should be considered as a baseline.
* Can staggered asynchronous inference be applied to multi-agent systems?

**Questions:**

I don't have other questions.

---

> ### Author Response · Authors · 2024-11-28
>
> Thank you for your review our paper. We really appreciate your praise of our novel theoretical contributions and experiments. We have  attempted to address each of the comments you made in the weaknesses section of your review below:
>
> **Testing Strategies Other Than DQN:** As our main contribution is focused on policy inference, we only implement DQN policy learning for our method and all baselines because it is a very generic choice for the learning method. We were wondering if you could elaborate a bit more on this point. Do you have any specific algorithms in mind that you would like to see us try? We are trying to get a sense of what questions you feel are currently unanswered regarding the choice of only considering the DQN for learning so that we can properly address this concern.
>
> **Parallel Updates Baseline:** Thank you for this helpful suggestion, it is indeed an interesting ablation to consider within the discourse of our paper. We have added the result of new experiments to address this comment in Figure 4a of the revised paper. For this experiment with the 100M parameter model, we had leveraged 33 learning processes with a batch size per process of 16 in order to keep up with the environment. As such, for the parallel update baseline, we accumulate gradients from each process and only update the policy parameters with the result of this big batch once all processes have been computed. This implies that the effective batch size is 33 times larger and that there are 33 times fewer parameter updates over the course of learning. The gradients from the larger batch size does not seem to be effective enough at integrating the data to make up for so many fewer updates to the model. This is in line with our discussion of the decision to consider asynchronous learning in Section 3.1. We have experiments for this ablation also running right now that we will add to Figure 4b. Unfortunately, we were not able to get access to the compute resources needed to get them done in time for the paper revision deadline, but we will definitely include this for the camera ready draft.
>
> **Application to Multi-agent Systems:** Staggered asynchronous inference can definitely be applied to multi-agent systems and we believe that this is a very interesting direction for future work. Based on your suggestion, we have added Appendix C in which we discuss the key challenges that should be considered to enable real-time multi-agent use cases in the future starting at line 1173.

---

> > ### Author Response · Authors · 2024-12-02
> >
> > **Rainbow Experiments:** We just wanted to follow up on our response because we have been running experiments over the weekend on the three Atari domains from Figure 6 with Rainbow rather than DQN used for the learning algorithm. We made this specific choice based on our discussion with Reviewer Pk39. Unfortunately, we were not able to get enough computational resources to fully complete these experiments before the end of the discussion period. However, we will definitely add them for the camera-ready version of our paper and our experiments for the sequential interaction baseline are almost entirely done now. What we find is that in the regime of low parameters counts, and thus low amount of delay and inaction, Rainbow performs a bit better for Boxing and Krull, and a bit worse for Name This Game. In the regime of high parameters counts, and thus high amounts of delay and inaction, sequential interaction reaches virtually the same performance in each game regardless of the architecture depth or learning algorithm. This makes sense as the rate of inaction is so high in this case that there is little that can be accomplished by learning that will change overall performance (with all variants converging to the performance of the all noop policy for the 1B parameter model). Meanwhile, our preliminary results with asynchronous interaction indicate that the Rainbow experiments will show a consistent overall trend with the DQN experiments -- a small gap with sequential interaction for small model sizes that grows very large as the model size grows large.
> >
> > With the discussion period coming to an end, we also wanted to see if you still have any remaining questions or concerns about our paper that we have not yet addressed.

---

### Official Review · Reviewer_Pk39 · 2024-10-29

**Soundness:** 3
**Presentation:** 3
**Contribution:** 3
**Rating:** 6
**Confidence:** 4

**Summary:**

This paper addresses a key challenge in applying RL models in real-time environments: the need for rapid, continuous responses in settings where even minor delays can accumulate into significant regret. The authors analyze regret bounds and show that minimizing long-term regret is often impractical in conventional sequential RL settings but becomes achievable with sufficient asynchronous computing resources. They introduce new algorithms to stagger inference processes asynchronously, ensuring that actions can still be made consistently, even with larger models. Empirical results on Game Boy games like Pokémon and Tetris support these theoretical insights, showing that the proposed methods support much larger models than previously feasible in real-time environments. The findings suggest that performance relies more on the environment's predictability over the inference period than on rapid action frequency, making it possible to deploy large-scale models in real-time.

**Strengths:**

The paper presents an advancement in asynchronous interaction and learning within real-time RL, addressing the challenge of deploying large neural network models in environments where timing is crucial.

The authors define an induced delayed semi-MDP which connects the asynchronous nature of real-time environments to the traditional sequential interaction paradigm. This framework allows for a deeper analysis of the sub-optimality that arises from time discretization and computational delays. Furthermore, they propose a novel decomposition of real-time regret into three components: regret from learning, regret from inaction, and regret from delay. This decomposition sheds light on the limitations of sequential interaction and the advantages offered by asynchronous approaches. A notable innovation in the paper is the introduction of algorithms for staggered asynchronous inference, which enable the use of models with longer inference times without compromising the frequency of actions. This key development facilitates the deployment of larger models in time-sensitive settings.

In summary, this paper makes some contributions to the field of real-time RL. The authors present a novel problem formulation, and provide some empirical evidence to support their findings. The presentation and detailed information enhance reproducibility. Overall, the work has important implications for the deployment of large models in real-time settings and may lead to advancements in RL software and hardware design.

**Weaknesses:**

I have some questions and concerns about this work that I provide in the following.

- In Section 2, it is mentioned that the time between decisions only depends on the MDP. How does it **only** depend on the MDP? Since it denotes the time between decisions, doesn’t it depend on policy instead? The respective sentence doesn’t make sense.
- About taking actions from the behavior policy:
    - How to select the default behavior that is safe and performant at different time steps? What if the default behavior policy leads the MDP_async into OOD states wrt the learned policy?
    - How can we make sure that the behavior policy mimics the real world as it evolves, considering factors such as changing environmental dynamics, varying agent interactions, and the need for continual adaptation to new scenarios?
    - If we change noop with the worst or the best action, how the results and conclusions would change? How important it is to the core idea of the paper?

- In the Empirical Results Section, Questions 1 and 2 are not hyperlinked while Questions 3 and 4 are. I’d suggest making them consistent.
- The experimental section raises two slightly different questions about the performance of asynchronous interaction and learning in a real-time strategy (RTS) game with constant novelty (lines 346 and 424). The first question, stated in the general description, emphasizes evaluating the “speed of progress,” “action throughput,” and “maintaining learning performance.” In contrast, the experiment-specific question appears to narrow the focus to determining whether asynchronous methods can indeed enable faster progress despite longer model and inference times. While both questions relate to asynchronous methods' impact on learning and progression, their phrasing suggests slightly different evaluation criteria, which may confuse readers about the primary aim of the experiments. The first question frames the experiments more broadly, implying a focus on learning stability and throughput, whereas the second centers specifically on asynchronous methods' ability to handle extended inference times in a constantly learning environment.

- About lines 433 and 434: Does the performance improvement over sequential interaction need to be statistically significant? If yes, then how are the results in Figs. 9 and 10 justified when the improvement is only significant for |\theta|=100M? If no, then it would be helpful to include a discussion explaining why statistical significance is not required to interpret these results as meaningful. For instance, if the improvement trends observed with smaller |\theta| values still support the theoretical advantages outlined in Remarks 1 and 2, an explanation should clarify why these trends are practically relevant, even if not statistically significant. Such clarification would strengthen the interpretation of the scaling benefits claimed for asynchronous methods.
- Why are confidence intervals omitted from Figs. 7 and 8?
- The empirical evaluation is only conducted with DQN as the baseline method. To strengthen the evaluation, consider including more advanced baselines like QR-DQN as well. Comparing with such variants would provide a more comprehensive view of the asynchronous framework's performance across different reinforcement learning strategies.

**Questions:**

See weaknesses.

---

> ### Author Response · Authors · 2024-11-28
>
> Thank you for taking the time to provide a detailed review of our paper. We really appreciate the way you highlighted our novel contributions and potential impact on the field. We have attempted to address each of the questions and concerns that you raised in your review below.
>
> **Only Depends on the MDP (Start of Section 2):** Thank you for pointing out how confusing this phrasing was in our submission. We were definitely not trying to say this is the case for the asynchronous interaction framework. Rather, what we were attempting to convey is that this is an unrealistic implicit assumption in papers that consider the standard RL interaction framework. This is the exact opposite of the case for the asynchronous interaction setting we consider in this work. Indeed, the entire point of introducing the concept of the induced delayed semi-MDP is because the time between decisions is directly controlled by the agent and we would like to thus understand the tradeoffs associated with particular choices. We have updated our discussion of this in the first paragraph of Section 2 (starting at Iine 81) accordingly.
>
> **Selecting the Default Behavior Policy:** Thank you for bringing up these interesting questions regarding the design of the default behavior policy. In our work, we assume this policy is provided as part of the environment and in fact our theory considers the worst case scenario in which following it always leads to suboptimality. As such, our primary focus in this paper is on avoiding use of it altogether rather than trying to design it to be a safer option. However, we believe that exploring these questions in more detail would be a very interesting avenue for follow up work inspired by the formulation of our paper. We have added a discussion of challenges and avenues for future research related to designing safe default behavior policies in Appendix C.
>
> **Impact of Swapping Noop with the Worst or Best Action:** Our worst case bound for inaction regret in Equation 3 of Theorem 1 relies on the idea that the default policy always follows suboptimal or worst case behavior. If the default policy always followed the best action, the optimal policy would be to never act in the environment, which is a scenario of limited interest to realtime RL scenarios. It is important to point out that changing the default behavior would have no impact on the results of our method using asynchronous inference, as we supply enough inference processes to act at every step and thus never follow the default behavior policy. For the sequential interaction baseline, whether other action choices would help depends on the nature of the game. For example, random actions would help with progress through episodes but lead to less success within those episodes than “noop” actions as i.e. in Pokemon sometimes actions cannot be reverted and guarantee failure after they are taken. It is a nice property of “noop” actions in Pokemon that they never have irrevertible consequences, which is a major difference with the Tetris scenario.
>
> **Questions 1 and 2 are not hyperlinked while Questions 3 and 4 are:** We really would like to address this when we update the paper, but feel like there must be a miscommunication. We re-downloaded our submitted draft on different computers and opened it with different applications. There is no hyperlink on any of the questions and the formatting is consistent in latex. Are you suggesting that we should add hyperlinks? Are you using a particular software to open it? Is it consistent for you in the updated draft or is there still an issue?
>
> **Comparing Statements of Question 1:** Thank you for pointing out that this was confusing. We conflated these concepts because from our perspective achieving higher action throughput and extended inference times are two sides of the same coin as the action throughput is equal to the inference time divided by the number of staggered asynchronous inference threads for our approach. So the point we were trying to make is that better performance in these environments is not simply achievable with higher throughput and also requires the agent to learn effectively to move on. We have rephrased the two versions of the question now so as to avoid confusion about consistency of the meaning.

---

> > ### Author Response · Authors · 2024-11-28
> >
> > **Statistical Significance at 1M and 10M:** Thank you for mentioning your confusion related to this result. Our theory does indeed demonstrate that the difference between asynchronous interaction and sequential interaction is expected to be less when the action inference time is low. Our results were only for 3 seeds in the experiment and we will add more for the camera ready version so we can be more confident about our conclusions. In this experiment, the sequential interaction baseline acts every 5 steps at 1M parameters, 8 steps at 10M parameters, and 70 steps at 100M parameters. For the Pokemon game it is expected that there is not much improvement acting at every step as it is well known to speed runners that in common circumstances the game could take as many as 17 frames to respond to non-noop actions in which time intermediate actions are "buffered" and not yet registered in the environment. As a result, these results are in line with expectations. We have added further discussion on this topic to Appendix A.1 to contextualize these findings. The impact of infrequent actions on performance is quite game specific. For example, in Tetris we see that even small degrees of inaction can be quite catastrophic. On the other hand, the new Atari experiments we added to Figure 6 in the revision reveal that performance gains are also modest at 1M and 10M for Boxing and Krull, but much bigger for Name This Game. We consistently see a very big difference as the model size gets large, but when that difference kicks in depends a lot on the specifics of the game.
> >
> > **Confidence Intervals for Figure 7 and 8:** Thank you for bringing this up. We excluded confidence intervals from the submitted draft just because run to run variance is so low when using consistent hardware that we were worried about even being able to see it on a chart. We have added confidence intervals to Figures 7 and 8, trying to also adjust the format to make them as clear to see as possible.
> >
> > **Testing Strategies Other Than DQN:** As our main contribution is focussed on policy inference, we only implement DQN policy learning for our method and all baselines because it is a very generic choice for the learning method. We did just want to clarify this based on your wording, as DQN learning is employed by all methods, including both our asynchronous inference algorithms and the sequential interaction baselines. We were wondering if you could elaborate a bit more on this point. Was there a specific reason you mentioned the QR-DQN model? Or is this just a proxy for any state of the art level method? We were considering if something like Rainbow or PPO would also address your comment. We just weren’t clear on what specifically you would be looking to see in this experiment as our method is agnostic to the choice of learning algorithm. If part of your comment was just trying to see more breadth in terms of results, we should point out that we added experiments in Figure 6 of our revision further corroborating our findings on realtime simulations of three Atari games, including ablations with respect to the number of layers in the ResNet architecture.

---

> > > ### Comment · Reviewer_Pk39 · 2024-11-29
> > > **Reviewer Response**
> > >
> > > I appreciate the authors' efforts in addressing my questions and concerns about the work.
> > >
> > > Regarding the hyperlink issue, it appears consistent now. I am not suggesting whether hyperlinks should be included or not, as that remains your decision.
> > >
> > > Concerning the use of DQN as the policy learning method, my suggestion to consider QR-DQN was intended merely as an alternative to DQN, primarily for comparison with the provided results. As the authors have noted, other algorithms such as PPO or Rainbow would also be suitable choices.

---

> > > > ### Author Response · Authors · 2024-12-02
> > > > **Re: Reviewer Response**
> > > >
> > > > **Rainbow Experiments:** Thank you for clarifying your concern. Based on your response, we have been running experiments over the weekend on the three Atari domains from Figure 6 with Rainbow rather than DQN used for the learning algorithm. Unfortunately, we were not able to get enough computational resources to fully complete these experiments before the end of the discussion period. However, we will definitely add them for the camera-ready version of our paper and our experiments for the sequential interaction baseline are almost entirely done now. What we find is that in the regime of low parameters counts, and thus low amount of delay and inaction, Rainbow performs a bit better for Boxing and Krull, and a bit worse for Name This Game. In the regime of high parameters counts, and thus high amounts of delay and inaction, sequential interaction reaches virtually the same performance in each game regardless of the architecture depth or learning algorithm. This makes sense as the rate of inaction is so high in this case that there is little that can be accomplished by learning that will change overall performance (with all variants converging to the performance of the all noop policy for the 1B parameter model). Meanwhile, our preliminary results with asynchronous interaction indicate that the Rainbow experiments will show a consistent overall trend with the DQN experiments -- a small gap with sequential interaction for small model sizes that grows very large as the model size grows large.
> > > >
> > > > With the discussion period coming to an end, we wanted to see if you still have any remaining concerns about our paper that we have not yet addressed. Has your opinion about the paper changed at all based on our responses?

---

### Official Review · Reviewer_NeQu · 2024-10-30

**Soundness:** 3
**Presentation:** 3
**Contribution:** 3
**Rating:** 8
**Confidence:** 2

**Summary:**

This paper considers real-time reinforcement learning where the environment continually evolves as the agent performs inference, thereby introducing potential delays on various aspects. The paper proposes a regret decomposition into three terms---learning time, inaction time (i.e. default behaviour while policy is performing inference), and irreducible regret due to stochasticity of the environment with delays. The paper focuses on reducing the two former terms through asynchronous learning and inference. The paper conducts experiments on three environments with varying model sizes. The experiments demonstrate a positive (negative) correlation with model size and delays (performance) with the sequential formulation, and further demonstrate that the proposed asynchronous variant can reduce the performance degradation.

**Strengths:**

- The paper is mostly well written and the decomposition seems reasonable.
- The proposed method is simple and easily applicable to real-life systems.
- The experiments seem mostly convincing

**Weaknesses:**

- I would appreciate if the paper considers $\Delta_{delay}$ by artificially injecting noise to the transitions to see the theoretically described bound.
- While the decomposition is nice, I am also interested in seeing whether the regret bound is sublinear. I was wondering if the authors have considered theoretically quantifying these algorithms under this framework.

**Questions:**

- I believe the following questions are captured by $\Delta_{delay}$, please correct me if I am wrong:
	- How does technique like action chunking play into this? In those situations I believe $\beta$ will be the actions taken by the policy from state $s$ which will be sufficient with more deterministic environments?
	- Furthermore, how meaningful is the asynchronous inference if the inference time is too coarse such that $\pi(s_{t-1}) \neq \pi(s_t)$ assuming no inference delays?
- I am unsure how to interpret Figure 8, particularly why is $N^*_\mathcal{L}$ larger with smaller batch size? I would expect the inverse as it should be less time-consuming to update smaller batch sizes?

---

> ### Author Response · Authors · 2024-11-28
>
> Thank you for your insightful review of our paper. We really appreciate you highlighting the strength of our writing, the applicability of our method for real-world systems, and the quality of our empirical results. We wanted to make sure that we addressed each of the questions and concerns you raised in our response.
>
> **Artificially Injecting Noise:** Thank you for suggesting this analysis. This is a very interesting idea. Unfortunately, we were not clear on how we could do this in a reasonable way for complex environments from our submission such as Pokemon and Tetris. If you have any suggestions on how this could be implemented, we would be happy to give it a try. We did, however, add some additional experiments using a realtime simulation of three Atari games in Figure 6 of our revised draft. One trend that we thought somewhat spoke to your idea is the different scaling behavior we see for asynchronous interaction on the games Boxing and Krull. In both cases, we see much better performance than sequential interaction at large model sizes, but there is a big difference in the degree to which asynchronous methods can be scaled up while still achieving human-level performance. In Boxing, we see it is not possible to achieve human-level performance with even a small amount of delay, performing worse than human-level for anything with 3 or more steps of delay. Meanwhile, for Krull we find that it is possible to maintain human-level performance with as many as 750 steps of delay. This makes sense in the context of these games as the opponent has a very stochastic policy in Boxing whereas the dynamics of Krull are far more predictable.
>
> **Do Algorithms Achieve Sublinear Regret?:** Thank you for bringing up this constructive comment as well. It is a natural question arising from the result we demonstrate in Remark 2 that fits really well with our overall discourse. We have added a footnote about this surrounding Remark 2 in the main text and have added Section B.6 of the Appendix to discuss this point in detail. The main conclusion is that combining Algorithms 1 and 2 with Q-Learning that leverages appropriate optimism and learning rates can guarantee a regret upper bound that is sublinear in time for deterministic environments. More generally, regret will be sublinear in time as long as delay regret is sublinear in time.
>
> **Connection to Action Chunking:** This was another very insightful comment. As we have now clarified in Section 2, we are focussed on policies that produce one action at a time in this paper. However, the action chunking approach is indeed quite relevant as a point of comparison. We have added a footnote about it to the main text and have added Section B.5 of the Appendix to discuss the action chunking approach in detail. One takeaway we highlight is that while there may be some circumstances in which action chunking can be used to eliminate inaction, this is not the case in general (as we can demonstrate for staggered asynchronous inference in Remark 2). Our other key conclusion is that action chunking leads to an action space that grows exponentially with the growing chunk size and that this in-turn leads to an exponential increase in the lower bound on regret from learning.
>
> **The Case When $\pi(s_{t-1}) \neq \pi(s_{t})$:** We were not totally sure we understood the question, but one point to clarify is that our policies learn to accommodate for the lag in action execution. For example, if action inference takes $k$ steps, our asynchronous inference policies learn to choose $a_{t+k}$ appropriately given $s_{t}$ rather than $a_{t}$. As a result, it is possible to learn a very different $a_{t+k}$ than the $a_{t}$ that would be learned in the case of no delay. Please let us know whether this explanation addresses your confusion.
>
> **Larger $N_{\mathcal{L}}^∗$ Values with Smaller Batch Sizes:** To clarify, $N_{\mathcal{L}}^∗$ represents the amount of processes needed to learn from all incoming environment transitions. As a result, while smaller batch sizes certainly do take less time to compute, they do not quite take linearly less time to compute (due to decreased opportunity for parallelism) to make up for the fact that they are processing fewer items. To illustrate when we mean, for the 10M parameter model the update time for batch size of 8 is 1.898 seconds, the update time for a batch size of 16 is 3.234 seconds, and the update time for a batch size of 32 is 6.061 seconds. But if we consider the update time per transition, a batch size of 8 corresponds to 0.237 seconds, a batch size of 16 corresponds to 0.202 seconds, and a batch size of 32 corresponds to 0.189 seconds. As a result, more processes will be needed to keep up with the throughput of transitions from the environment with smaller batch sizes to accommodate for the greater amount of time needed to learn from a transition in each process. Please let us know if this still doesn’t make sense and we would be happy to clarify.

---

> > ### Comment · Reviewer_NeQu · 2024-12-01
> >
> > Thank you for the response. I believe you have answered my questions, including the case where $\pi(s_{t-1}) \neq \pi(s_t)$.
> >
> > Regarding artificial noise, what I was suggesting was a toy environment (e.g. $N$-state MDPs for some small $N$) for which we have control all distributions, including the delays, that would allow us to compare with the theoretical bounds. It's just to make sure if we follow the assumptions in theory we get similar results in experiments.

---

### Official Review · Reviewer_a7T1 · 2024-11-04

**Soundness:** 3
**Presentation:** 3
**Contribution:** 2
**Rating:** 6
**Confidence:** 2

**Summary:**

This paper addresses the challenges of employing large models for real-time reinforcement learning, where high inference latency prevents conventional sequential inference-and-learning processes from converging. The authors first formulate asynchronous interactive learning as a delayed semi-Markov Decision Process (semi-MDP) and analyze regret lower bounds by decomposing them into several theoretical components. Their analysis demonstrates that staggering multiple processes can enable effective learning while sequential learning cannot.

The paper then proposes two algorithms that stagger multiple processes for asynchronous RL by maintaining a constant action interval. On the evaluated game benchmarks of Pokemon Game Boy and Tetris, the proposed methods significantly outperform sequential methods, especially when the model size is large. Additionally, this paper analyzes the scaling properties of the number of inference processes needed.

**Strengths:**

- **Originality:** To the best of my knowledge, the decomposition of regret bounds into several components under real-time settings is interesting and novel. The proposed algorithms are well-motivated by this theoretical analysis.
- **Clarity:** The paper is well-written and easy to follow. The presentation is clear, including comprehensive details on problem formulation, theoretical proofs, and experimental settings with results.

**Weaknesses:**

- The paper's impact could be broadened by evaluating more experimental settings and environment domains to demonstrate the generality of the theory-motivated algorithms. For instance, including locomotion/control environments with continuous action spaces would provide stronger evidence for applying these methods to real-time robotics applications.
- On question 2: this paper could be more convincing, with more detailed discussions explaining why sequential learning methods fail as network size increases, for instance, including concrete examples demonstrating how large networks prioritize high-frequency actions could provide evidence to support the theoretical analysis empirically in a more grounded way.
- When comparing models of different scales, one potential extension is to consider variations in the number of layers, in addition to the number of channels, would provide a more comprehensive evaluation since both factors significantly affect inference time.
- Algorithm 2 would benefit from experiments/results in the settings with relatively high inference time variance. Though this is a minor point, such experiments would better justify the choice between two algorithms.

**Questions:**

Please see the above section.

---

> ### Author Response · Authors · 2024-11-28
>
> Thank you for your review of our paper. We really appreciate the way your review highlighted our theoretical contributions and the presentation of our paper. You also made some great suggestions in the weaknesses section that we wanted to make sure to address in our response and revisions.
>
> **Evaluating More Experimental Settings:** This is a great suggestion. We have added a new set of experiments to augment our experiments provided in the submitted draft in Figure 6 of our revision. We considered a realtime simulation of three Atari games (Boxing, Krull, and Name This Game) following a similar procedure to our simulation of the Game Boy. The particular games were selected to allow for learning an agent that performs on-par with humans within 2,000 episodes of training without the presence of delay. This way we can clearly see the degradation of performance as the degree of delay and inaction are increased with large models. Our experiments in these new domains further corroborate our key findings in Tetris and Pokemon, demonstrating the generality of our proposed approach.
>
> **Why sequential learning fails as network size increases:** Sequential interaction fails as the network size increases because more inaction and delay is experienced when the action inference time becomes larger.  For example, in a ResNet model with sequential interaction, a non-noop action is only taken every 5 steps for the 1M model, every 8 steps for the 10M model, every 70 steps for the 100M model and every 596 steps for the 1B model. In all of our experiments, we plot the performance of a policy that takes all noop actions and we see that larger models converge towards this behavior within the sequential interaction framework. Please let us know if this clarifies your concern. We were not really sure what kind of evidence you were additionally looking for in terms of qualitative examples of behavior.
>
> **Variations in the number of layers, in addition to the number of channels:** This was another great suggestion. We added an ablation to our new experiments in Figure 6 with a deeper variant of the ResNet architecture we consider leveraging 30 convolutional layers (as opposed to 15 convolutional layers). As a result, the channel number multiplier needed to get to the same number of parameters is lower and the inference time at the same number of parameters is higher. As such, there is even more delay and inaction for algorithms to deal with at a particular model size. Predictably, we see that the main effect is that the performance of the sequential interaction baseline falls off even faster.
>
> **Experiments with high inference time variance to validate Algorithm 2:** We actually did try to address this point in Figure 7 of the submitted draft, which we have updated for increased clarity. The idea is that although the neural network forward propagation time has low variance, this time is quite different from the potential speed of a simple random number generator during exploration steps. As such, we compare the throughput at $\epsilon=0$, where all actions are taken by the neural network and the variance in action inference times is low, with the throughput at $\epsilon=0.5$, when half of all actions are very fast random actions. In line with our theoretical predictions, we find that Algorithms 1 and 2 require the same number of total processes when the inference time variance is low and that Algorithm 2 is more efficient when the variance is high. We explain this starting at line 454 in the revised draft.
>
> Please let us know if you have any additional questions or concerns based on our explanation and we would be happy to clarify during the discussion period.

---

> > ### Author Response · Authors · 2024-12-02
> >
> > As the discussion period is coming to an end, we wanted to check in to see if we have addressed all of your concerns or if there are any questions you still have about the paper. Please let us know if there is anything you would like us to clarify further.

---

### Author Response · Authors · 2024-11-28
**General Response: Thank You For Your Reviews!**

We wanted to thank all of the reviewers for providing detailed and thoughtful reviews of our paper. We really appreciate the significant praise of our paper expressed by each reviewer. Specifically, we were really happy to see multiple reviewers highlighting the novelty and practicality of our theoretical contributions, the quality of our writing, and the significance of our experimental results. We have attempted to address every question and concern raised by each reviewer in the individual responses.

**Main Revisions to the Paper:** As multiple reviewers seemed focused on the impact of learning algorithms, we decided to slightly edit our title to further emphasize that our novel algorithmic contribution is specifically with respect to how inference is performed. We also added experiments based on the suggestions made by reviewers. In Figure 6 of the revised draft we have added experiments for realtime simulations of three Atari games as a function of two different architecture configurations. We have also added shading to highlight statistical significance in all figures (even when variance is very small) and a new parallel update baseline to Figure 4a. Moreover, we added some new sections to the appendix including Appendix B.5 to discuss action chunking approaches, Appendix B.6 to discuss when our algorithms provably achieve sublinear regret with respect to time, and Appendix C to discuss further considerations related to deployment for increasingly complex real-world use cases.

We look forward to continuing to engage with the reviewers during the discussion period.

---

### Meta-Review · Area_Chair_DwJr · 2024-12-29

**Metareview:**

This paper addresses the mismatch between discrete-time RL and real-world continuous time, focusing on the limitations imposed by action computation time, especially for large models in high-frequency environments. The authors formalized the regret decomposition and provides analysis, and proposed staggered asynchronous inference methods. Further empirical study has been conducted on real-time games like Pokémon and Tetris.


All the reviewers acknolwdge the novelty in regret decomposition and the soundness of the proposed algorithms. It is a clear acceptance.

**Additional Comments On Reviewer Discussion:**

The authors provides more empirical evidence and all the reviewers kept positive to the paper.

---

### Decision · Program_Chairs · 2025-01-22

Accept (Poster)